

# Importance of ice elasticity in simulating tide-induced grounding line variations along prograde bed slopes

Natalya Maslennikova[1], Pietro Milillo[1,2], Kalyana Nakshatrala[1], Roberto Ballarini[1], Aaron Stubblefield[3], Luigi Dini[4]

[1]Department of Civil & Environmental Engineering, University of Houston, TX, USA
[2]German Aerospace Center (DLR), Microwaves and Radar Institute, Munich, Germany
[3]Thayer School of Engineering, Dartmouth College, Hanover, NH, USA
[4]Italian Space Agency (ASI), Matera, Italy

*Correspondence to*: Natalya Maslennikova (nmaslenn@cougarnet.uh.edu)

**Abstract.** The grounding line, delineating the boundary where a grounded glacier goes afloat in ocean water, shifts in response to tidal cycles. Here we analyze COSMO-SkyMed Differential Interferometric Synthetic Aperture Radar data acquired in 2020 and 2021 over Totten, Moscow University, and Rennick glaciers in East Antarctica, detecting tide-induced grounding line position variations from 0.5 to 12.5 km along prograde slopes ranging from ~0 to 5%. Considering a glacier as a non-Newtonian fluid, we provide two-dimensional formulations of the viscous and viscoelastic short-term behavior of a glacier in partial frictional contact with the bedrock, and partially floating on sea water. Since the models' equations are not amenable to analytical treatment, numerical solutions are obtained using FEniCS, an open-source Python package. We establish the dependence of the grounding zone width on glacier thickness, bed slope, and glacier flow speed. The predictions of the viscoelastic model match ~93% of all the DInSAR grounding zone measurements and are ~71% more accurate than those of the viscous model. The results of this study underscore the critical role played by ice elasticity in continuum mechanics-based glacier models, and being validated with the DInSAR measurements, can be used in other studies on glaciers.

## 1. Introduction

The grounding line, which defines the boundary between the ice sheet (the portion of a glacier laying on the bedrock) and the ice shelf (the portion floating on the ocean water), is of particular significance for comprehensive Antarctic investigations (Friedl et al., 2020; Haseloff and Sergienko, 2018; Schoof, 2007a). The grounding line is a crucial indicator of glacier stability, as its position reflects the salient glacier dynamics and influences the overall glacier force and mass balances (Davis et al., 2023; Davison et al., 2023; Holland, 2008; Marsh et al., 2016). Grounding lines not only provide valuable information about glacier stability by enabling the evaluation of ice thickness, but also allow the monitoring of sea level changes due to climate warming (Schoof, 2007a; Goldstein et al., 1993; Friedl et al., 2020). The mechanism governing variations in grounding line position is complex and involves both long-term and short-term processes (Sergienko, 2022; Sergienko and Haseloff, 2023). Here we focus on short-term grounding line migrations induced by tidal forces and occurring within a tidal cycle (Coleman et al., 2002; Dempsey et al., 2021; Albrecht et al., 2006; Freer et al., 2023; Beldon and Mitchell, 2010; Warburton et al., 2020). Differential Interferometric Synthetic Aperture Radar (DInSAR) and altimeter techniques applied across various Antarctic glaciers have shown that the magnitude of tide-induced grounding line migrations can extend to several kilometers; several orders of magnitude wider than the grounding zone width expected from hydrostatic equilibrium (Brancato et al.,



2020; Freer et al., 2023; Minchew et al., 2017; Begeman et al., 2020; Milillo et al., 2022; Dawson and Bamber, 2017;
Brunt et al., 2010). Long-term glacier models primarily aim to estimate grounding line evolution over time scales
significantly exceeding tidal scales, thus neglecting short-term variations (Thomas and Bentley, 1978; MacAyeal,
1989; Schoof, 2007b; Weertman, 1974; Gudmundsson et al., 2012; Durand et al., 2009b, a; Cornford et al., 2020;
Favier et al., 2014; Gagliardini et al., 2016; Pattyn et al., 2012; Seroussi et al., 2014; Muszynski and Birchfield, 1987;
Pegler and Worster, 2013; Pegler et al., 2013; Robison et al., 2010). Conversely, short-term glacier models focus on
tidal time scales and tend to disregard the long-term evolution of glaciers due to its negligible impact over these shorter
periods (Rosier et al., 2014; Rosier and Gudmundsson, 2020; Stubblefield et al., 2021).
Short-term glacier dynamics has been studied using different physical approaches (Warburton et al., 2020;
Sayag and Worster, 2013, 2011; Tsai and Gudmundsson, 2015; Rosier and Gudmundsson, 2020; Rosier et al., 2014;
Stubblefield et al., 2021). The goal is to quantify the grounding zone width, or the amplitude of the tide-induced
grounding line migrations, by calculating the difference between the grounding line positions at high and low tides
(Ciracì et al., 2023; Gadi et al., 2023; Milillo et al., 2017, 2022; Chen et al., 2023a). For example, a hydrological
model proposed by (Warburton et al., 2020) defines the grounding zone width as the penetration depth into a subglacial
cavity of water interacting with an elastic ice beam that responds to the ocean tides. Several models consider a glacier
as a solid beam moving in a vertical direction due to the periodic tidal impact (Sayag & Worster, 2011, 2013, Tsai &
Gudmundsson, 2015, Chen et al., 2023). In (Sayag and Worster, 2013, 2011) a grounding line migration is considered
a result of the tidal force-induced deformation of an (elastic) Euler-Bernoulli beam. However, they treat the Young's
modulus of ice as a tidal phase-dependent model parameter to support the sustainability of the beam model in fitting
the satellite observations (Sayag & Worster 2013). (Tsai and Gudmundsson, 2015) consider grounding zones as an
opening and closing crack between an elastic ice beam and the bedrock, using equations governing the propagation of
a water-filled crack under pressure. This model, which cannot predict grounding line migrations at low tides, was
modified and applied to the Amery Ice Shelf in Antarctica by (Chen et al., 2023b), who showed that the crack model
can reproduce a kilometer grounding line retreat over a tidal cycle. Nevertheless, the crack-based method is one-
dimensional, as it takes into account only the glacier motion along the ice-bedrock surface and does not describe
motion-indued changes inside the ice. Other models treat glacier ice as a viscous or viscoelastic fluid and seek to
determine grounding line migration by resolving contact forces at the base (Rosier and Gudmundsson, 2020; Rosier
et al., 2014; Stubblefield et al., 2021). (Rosier et al., 2014) and (Rosier and Gudmundsson, 2020) designed nonlinear
viscoelastic models on tidal time scales, where the normal stress and velocity determine the grounding line position
but being considered after discretization, they are not included into the variational formulation used. This technical
detail was addressed by (Stubblefield et al., 2021), who used the full Stokes equations for purely viscous flow and
included contact conditions in the variational formulation.
DInSAR data acquired between 2020 and 2021 for Totten, Moscow University, and Rennick glaciers, located
in East Antarctica, show tidally induced grounding zone width values in the kilometer range, which exceed those
estimated from hydrostatic equilibrium by at least one order of magnitude (Brancato et al., 2020; Freer et al., 2023;
Milillo et al., 2022). To model these observations, we present a framework that accurately predicts the temporal
evolution of the grounding line using a continuum mechanics-based approach. We extend the viscous model, proposed



by (Stubblefield et al., 2021), by incorporating an elastic component within the framework of the upper-convected
Maxwell model (Snoeijer et al., 2020; Gudmundsson, 2011). The formulation involves variational inequalities
associated with the bedrock boundary conditions for water pressure, ice velocity and normal stress. The variational
inequalities are transformed into variational equalities via a penalty method. Grounding line migrations are obtained
by solving these variational equations at each model timestep using the Finite Element Method (FEM) in the open-
source FEniCS package (Alnæs et al., 2015; Logg et al., 2012). Glacier thickness, bedrock slope, and ice flow serve
as model inputs, and were set based on those for Totten, Moscow University, and Rennick glaciers using Bed Machine
(Morlighem et al., 2017) and Measures InSAR Version 2 (Rignot et al., 2017). Performing the comparison of
automatically generated grounding zones with the grounding zone width values, manually assessed from the DInSAR
interferograms, we evaluate both models' performance and assess the significance of the elastic component relative to
the formulation that accounts for only viscosity. Additionally, we determine the impact the ice-bed system's main
parameters, namely, bedrock slope, glacier thickness, and ice velocity, on the magnitude of tidally induced grounding
line migrations.

## 2.   Data and Methods

We assess the models' performance using grounding zone, glacier thickness, bedrock slope, and ice flow velocity
values that characterize Rennick (REN), Moscow University (MU), and Totten (TOT) glaciers (Figure 1). MU and
TOT are neighboring glaciers, located on the Sabrina Coast in East Antarctica (Orsi and Webb, 2022; Fernandez et
al., 2018; Bensi et al., 2022). Together, the combined effect of these two glaciers may result in to up to a 5-meter sea
level rise, making them major contributors to sea level changes in East Antarctica (Mohajerani et al., 2018). Being
characterized by the highest outflow and thinning rate in East Antarctica, TOT also has the third-largest ice flux among
all Antarctic glaciers, following Pine Island and Thwaites glaciers (Roberts et al., 2018; Rignot and Thomas, 2002;
Pritchard et al., 2009). In contrast, MU exhibits relatively slow thinning rates (Mohajerani et al., 2018) and nearly half
the basal melt rate of TOT: 4.7 ± 0.8 m/yr vs 10.5 ± 0.7 m/yr between 2003 and 2008, respectively (Rignot et al.,
2013). Both glaciers are grounded below the sea level, making them potentially unstable and susceptible to collapse
(Van Achter et al., 2022; Aitken et al., 2016). REN, situated in Northern Victoria Land in East Antarctica, spans over
400 km along the flow and narrows from 80 km to 25 km across the flow (Mayewski et al., 1979; Meneghel et al.,
1999; Sturm and Carryer, 1970; Allen et al., 1985). Containing the sea-level equivalent of 11 cm in a form of ice,
REN is also grounded below the sea level and is experiencing rapid thinning due to intensive basal melt (Pritchard et
al., 2012; Rignot et al., 2019). REN's ice discharge has shown up to 20% amplification between 1999 and 2018 (Miles
et al., 2022). Despite exhibiting similar behavior to TOT and MU, REN retreats slower than most Antarctic glaciers,
rendering it relatively stable (Miles et al., 2022; Pritchard et al., 2012).
We obtain the grounding zone width values for TOT, MU, and REN utilizing a series of 1-day repeat pass Synthetic
Aperture Radar (SAR) data from the COSMO-SkyMed mission, which comprises a four-satellites constellation
equipped with synthetic aperture radars operating at X-band with a wavelength of 3.1 cm (Milillo et al., 2014). Each
of the four satellites has a 16-day repeat cycle, while our analysis focuses on the images collected by the second and
third satellites, capturing data over the same area with a 1-day interval.



The interferograms were generated from the STRIPMAP mode of the COSMO-SkyMed, using a set of $4 \times 5$
consecutive overlaying frames with an azimuth and range resolution of 3 meters and a $40 \times 40$ km swath. To eliminate
the topographic effect, the Copernicus digital elevation models (DEMs) were employed. To co-register the data and
achieve maximum phase coherence we used satellite orbits for coarse co-registration and used a pixel offsets approach
for fine co-registration. A multi-looking factor of 10 in both range and azimuth was used to achieve an interferogram
resolution of 30 m x 30 m. Two one-day interferometric pairs were combined into one double differential
interferogram (DInSAR) to cancel out horizontal deformation due to glacier flow. Each interferometric pair combined
in a DInSAR interferogram was acquired within 1.5 months over the same satellite track in order to minimize
horizontal velocity changes. An interferometric fringe corresponds to half a wavelength of surface displacement,
equivalent to 1.5 cm of satellite line of sight displacement per fringe for X-band or about 1.7 cm when projecting
deformation onto the vertical considering the satellite incidence angle. The grounding line can be manually delineated
as the most inner fringe in the grounded ice side. Therefore, the DInSAR technique provides information about vertical
tide-induced glacier movements and enables grounding line mapping with accuracy of the order of 100 - 200 m
(Rignot et al., 2014).
We use three pairs of DInSAR interferograms, acquired between 2020 and 2021, where each pair covers the main
trunk of one of the glaciers of interest and is obtained within a 1.5-month period to ensure that any variations in the
grounding line position occur due to the tidal interaction rather than glacier retreat (Milillo et al., 2022). Combining
two manually mapped grounding lines for each interferogram in a pair, we establish a grounding zone for the
corresponding glacier and measure the grounding zone width along the ice flow directions.
To determine the average thickness of the glaciers and their corresponding bed slopes, the BedMachine Antarctica
(version 2) was employed (Morlighem et al., 2017). Additionally, the MEaSUREs (version 2) InSAR-based ice
velocity map of Antarctica was utilized in calculating the average ice flow speed (Rignot et al., 2017). All
measurements were performed along a total of 80 velocity flow lines, located 500 m to 600 m apart, each
approximately 20 km in length, a parameter adopted as the glacier domain length during the modeling process.



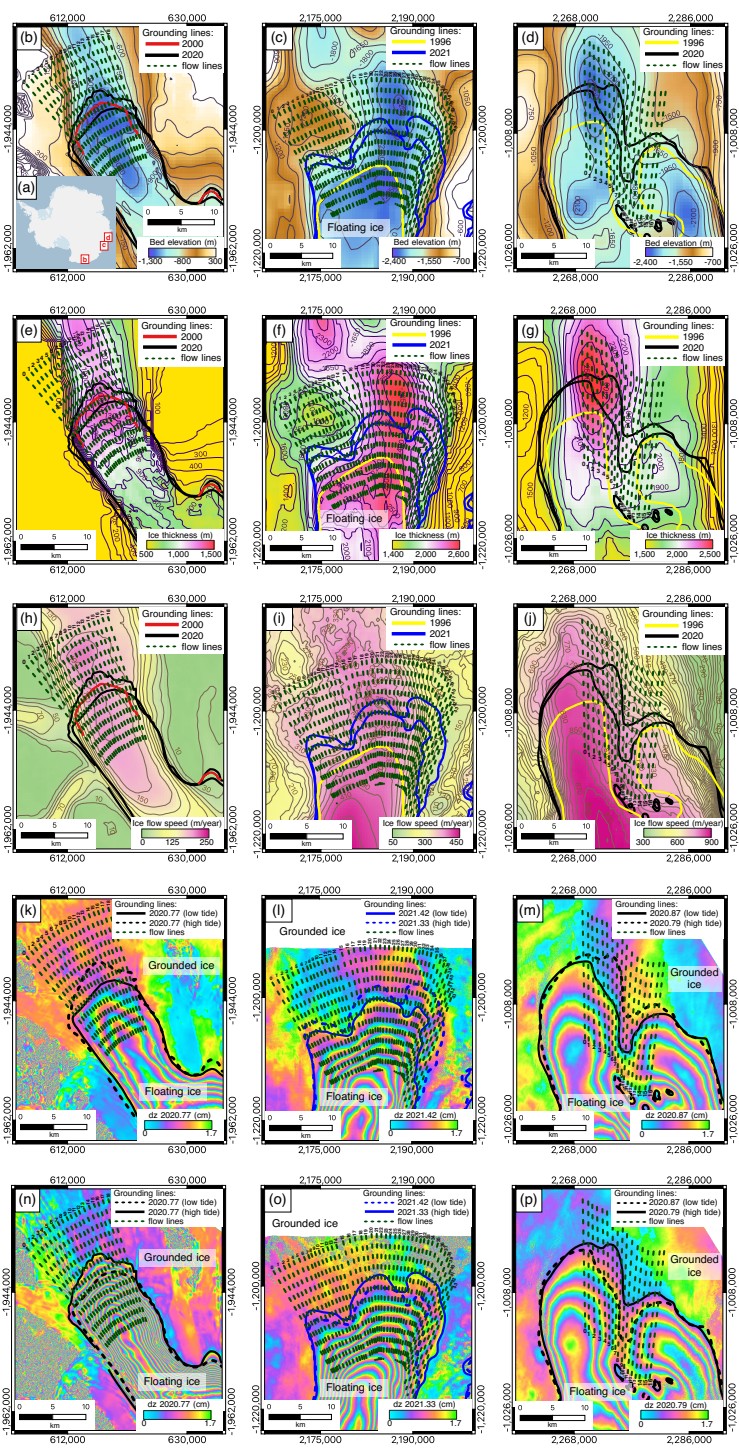



**Figure 1.** Study area: (a) location of Totten (TOT), Moscow University (MU), and Rennick (REN) glaciers in Antarctica. Subplots (b), (c), (d) show the bed elevation relief from BedMachine2 (Morlighem et al., 2017) with 50 m contour levels (purple) over REN, MU, and TOT, respectively. Subplots (e), (f), (g) show the ice thickness map with 100 m contour levels (purple) from BedMachine2 (Morlighem et al., 2017) over REN, MU, and TOT, respectively. Subplots (h), (i), (j) show the ice flow velocity map with 20 m/year contour levels (purple) from MEaSUREs2 (Rignot et al., 2017) over REN, MU, and TOT, respectively. Subplots (k), (l), (m) show the DInSAR interferogram at low tide with a corresponding grounding line as a solid line and a grounding line at high tide as a dashed line over REN, MU, and TOT, respectively. Subplots (n), (o), (p) show the DInSAR interferogram at high tide with a corresponding grounding line as a solid line and a grounding line at low tide as a dashed line over REN, MU, and TOT, respectively. The grounding lines over REN and TOT were mapped in 2020 (black line), while the grounding lines over MU (blue line) correspond to 2021. The grounding lines for 1996 (yellow line) and 2000 (red line) on figures (b) – (j) were taken from MEaSUREs2 DInSAR-based Antarctic grounding line dataset (Rignot et al., 2016). Numbered dark green dotted lines represent the flow lines, along which the measurements (Table S1) were performed. All datasets are represented in Antarctic projection (EPSG:3031).

### 3.  Viscous and viscoelastic models

The short-term grounding line migration model, rooted in the Navier-Stokes equations under the assumption of viscoelastic ice flow, is based on the purely viscous formulation of the same problem (Stubblefield et al., 2021). Here we summarize the similarities and differences between the models, while the comprehensive description of the viscoelastic model including the derivation of the model's penalized problem (A35), and the details about the models comparison are provided in Appendix A: Glacier modelling.

For both models, we designate the glacier domain as $\Omega$, as shown in Figure 2, with a piece-wise smooth boundary $\partial\Omega$. We place the glacier in the two-dimensional coordinate system $(X, Y)$, where $X$ denotes the horizontal axis, and $Y$ is used for identifying the vertical axis. In the principal notation used in this paper, a spatial point is denoted by $\boldsymbol{x} = (x, y) \in \bar{\Omega}$, where an overline denotes the set closure. For glacier length $L$, the ice domain $\Omega$ can be mathematically expressed as

$$\Omega = \left\{ (x, y): \ |x| < \frac{L}{2}, \ \ s(x,t) < y < h(x,t) \right\} \tag{1}$$

The glacier boundary is represented as a union of five complementary parts:

$$\partial\Omega = \Gamma_D \cup \Gamma_N \cup \Gamma_w \cup \Gamma_b \cup \Gamma_a, \tag{2}$$

where $\Gamma_D$ is an inflow boundary, $\Gamma_N$ is an outflow boundary, $\Gamma_w$ is an ice–water surface, $\Gamma_b$ is an ice–bedrock surface, and $\Gamma_a$ is an ice–air surface. Defining $f(x)$ as a bedrock slope function, $h(x, t)$ as a time-dependent function of the glacier surface elevation, and $s(x, t)$ as a function, defining the position of lower boundary of the ice shelf with time, the ice–water and ice–bed boundaries are expressed as

$$\Gamma_w = \{ (x, y) \in \partial\Omega: y = s(x,t) > f(x) \}, \tag{3}$$

$$\Gamma_b = \{ (x, y) \in \partial\Omega: y = s(x,t) = f(x) \}. \tag{4}$$

The entire lower boundary $\Gamma_s$ therefore, is identified as a union of the ice–water and ice–bed boundaries:



$$\Gamma_s = \Gamma_w \cup \Gamma_b = \{(x,y) \in \partial\Omega : y = s(x,t) \geq f(x)\} \tag{5}$$

The grounding line position is identified as a point, where the ice–water boundary intersects the ice–bed boundary.

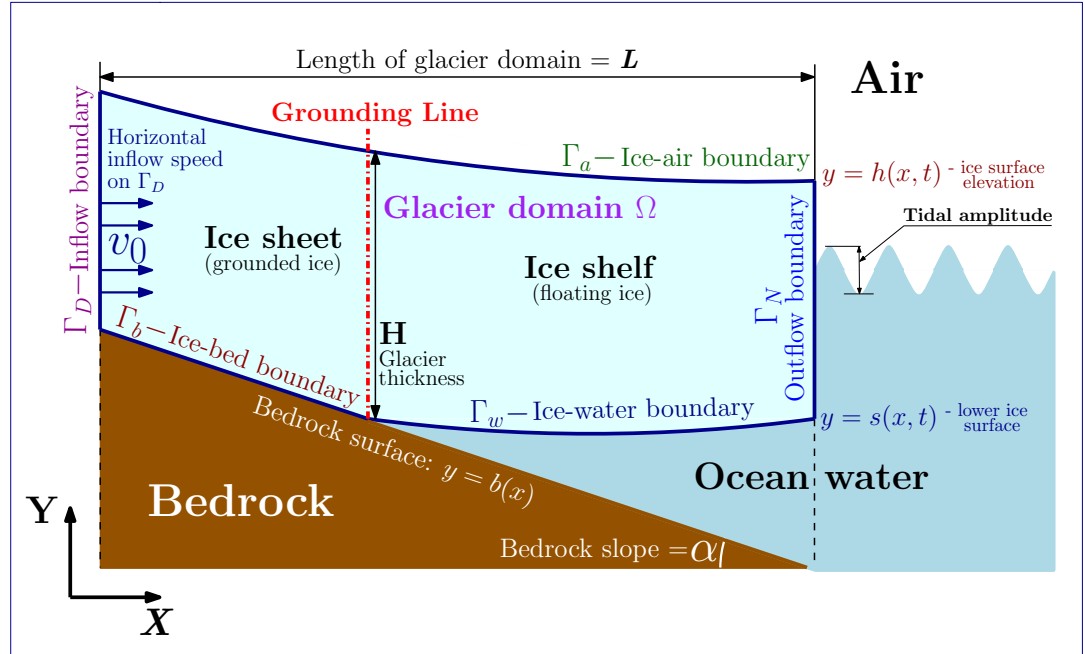

**Figure 2.** Model geometry for both viscous and viscoelastic problems, highlighting the ice domain $\Omega$ boundaries, as defined in Table A1.

The bedrock slope in the models is set using the function

$$b(x) = -\frac{Ax}{L}, \tag{6}$$

Where $A$ is a variable parameter in meters that determines the bedrock inclination, and $L$ is a glacier domain length, which is kept constant at 20 km for all model runs to ensure consistent results. The bed slope $\alpha$ is determined as the tangent of the bedrock function $b(x)$ and is measured

$$\alpha = \frac{A \cdot 100\%}{L}. \tag{7}$$

Both viscous and viscoelastic models require bed slope, glacier thickness and ice inflow speed as input parameters. For one set of input parameters, the code solves the corresponding variational problem twice: first, for a calm ocean surface without tides to stabilize the glacier and approximate its shape to a more natural geometry than the initially specified one; and second, for the tidal situation where the grounding zone width is determined. We employ sinusoidal-shaped tides with a 1 m amplitude and a half-day period $P$, which is typical for the investigated glaciers (Padman et al., 2018; Hibbins et al., 2010). Thus, the sea level in a tidal case changes with time as

$$l(t) = \frac{\rho_i}{\rho_w}H + sin\left(\frac{2\pi t}{P}\right), \tag{8}$$

where $H$ is the glacier thickness at the grounding line.



The grounding line position is defined based on the numerical tolerance $\xi$, set to 1 mm. If the computed position of a
lower boundary mesh node s is $\xi$ mm greater in the vertical direction than the bedrock, that node is classified as
floating. Conversely, if a node position does not deviate from the bed by more than $\xi$, that node is classified as
grounded. Schematically, the node classification can be described as:

$$\begin{cases} s - b \le \xi \implies \text{grounded node} \\ s - b > \xi \implies \text{floating node} \end{cases}.$$
(9)

Both models consider a glacier as an incompressible and non-Newtonian ice flow, sharing the same domain and
restricted by identical boundary conditions. Using FEniCS, a freely available FEM Python package, the models
employ Taylor–Hood elements for velocity and pressure fields to solve a corresponding variational problem on each
time step by means of a Newton solver for nonlinear systems of equations. The primary distinction between the viscous
and viscoelastic models lies in the incorporation of an elastic component, represented by Hooke's law. While the
addition of the elastic component enables the viscoelastic model to account for significant short-term glacier
deformations, as provided by the application of the upper-convected Maxwell model of viscoelasticity (see
A2.1. Governing equations), it also entails a substantial increase in computational resources required for a single
model run. A comparative analysis of the main parameters of the models is presented in Table 1. The detailed
exposition of the viscoelastic model is provided in Appendix A: Glacier modeling, while the main specifications of
the viscous model can be found in (Stubblefield et al., 2021).
**Table 1.** Comparison of the main properties of the viscous and viscoelastic models.

| Characteristics | Viscous model | Viscoelastic model | Similarity |
|---|---|---|---|
| **Material properties of glacier flow** | | | |
| Compressibility | Incompressible | Incompressible | Same |
| Rheological behavior | Non-Newtonian | Non-Newtonian | Same |
| Material behavior | Viscous | Viscoelastic | Different |
| **Physical formulation of the models** | | | |
| Glacier domain | Equations (1) – (5) | Equations (1) – (5) | Same |
| Boundary conditions | Equations (A17) – (A22) | Equations (A17) – (A22) | Same |
| *Governing equation* — Conservation of mass in case of incompressibility | Equation (A1) | Equation (A1) | Same |
| Conservation of momentum (Stokes equation) | Equation (A2) | Equation (A2) | Same |
| Constitutive law (Hooke's law) | Both Hooke's law (A3) and ice viscosity (A5) are defined via strain rate tensor (A4) | Both Hooke's law (A6) and ice viscosity (A8) are defined via deviatoric stress tensor $\tau$ (A7), which, in turn, depends on strain rate tensor (A4) | Different |
| **Implementation of the models** | | | |
| Penalized problem solved by the model on each time step | Equation (3.21) in (Stubblefield et al., 2021) | Equation (A35) | Different |



| Computation time*<br>(HH:MM:SS) | | | |
|---|---|---|---|
| Glacier thickness = 1.0 km | 01:07:07 | 02:57:48 | Different |
| Glacier thickness = 1.5 km | 01:34:29 | 04:21:10 | Different |
| Glacier thickness = 2.0 km | 01:59:54 | 06:07:50 | Different |
| Glacier thickness = 2.5 km | 02:41:39 | 07:30:08 | Different |

\* For 2.3 GHz 8-Core Intel Core i9 processor. Model input parameters: Domain length = 20 km; Bed slope = 1.0 %; Inflow speed = 100 m/year; Mesh size = 250 m and 50 m on upper and lower domain boundaries, respectively. The time is provided the HH:MM:SS format, where H shows the number of hours, M shows the number of minutes, and S is the number of seconds

## 4. Results

A total of 80 flow lines were utilized for computing ice-bed system parameters, namely, grounding zone width, ice thickness, bed slope, and ice flow speed at the grounding zone. All measured parameters are listed in Table S1, while a statistical summary of these measurements is provided in Table 2. The three glaciers rest on prograde bedrock slopes, gradually ascending inland. TOT exhibits the shallowest average bed slope among the glaciers of interest, measuring $1.1 \pm 0.1$ %. The glacier has average grounding zone width of $4.0 \pm 0.4$ km and a mean thickness of $2.1 \pm 0.1$ km, making it the fastest one among the three glaciers with an average speed of $691 \pm 77$ m/year. In contrast, REN is the thinnest and slowest among the three, with a mean thickness of $1.0 \pm 0.2$ km and a flow speed of $170 \pm 16$ m/year. It also features the smallest average grounding zone width of $2.3 \pm 0.4$ km and a rising inland bed with an average rate of $1.2 \pm 0.1$ %. MU, characterized by the largest mean grounding zone of $4.9 \pm 0.4$ km, also has the steepest average bed slope of $1.4 \pm 0.2$ %. With an average thickness of $2.2 \pm 0.1$ km, the glacier maintains a mean ice flow speed of $239 \pm 26$ %.

**Table 2.** Minimum, maximum, and average values of the grounding zone width, ice thicknesses, bed slopes, and ice flow speed of Totten (TOT), Moscow University (MU), and Rennick (REN) glaciers.

| Glacier characteristics | | TOT | MU | REN | Data source |
|---|---|---|---|---|---|
| **Grounding zone, km** | Min | 0.9±0.4 | 0.4±0.4 | 0.8±0.4 | Pairs of DInSAR interferograms |
| | Mean | 4.0±0.4 | 4.9±0.4 | 2.3±0.4 | |
| | Max | 7.6±0.4 | 12.5±0.4 | 3.3±0.4 | |
| **Ice thickness, km** | Min | 1.8±0.2 | 1.7±0.1 | 0.7±0.4 | BedMachine2 (Morlighem et al., 2017) |
| | Mean | 2.1±0.1 | 2.2±0.1 | 1.0±0.2 | |
| | Max | 2.4±0.1 | 2.4±0.2 | 1.2±0.1 | |
| **Bed slope, %** | Min | 0.02±0.02 | 0.01±0.01 | 0.3±0.1 | BedMachine2 (Morlighem et al., 2017) |
| | Mean | 1.1±0.1 | 1.4±0.2 | 1.2±0.1 | |
| | Max | 4.6±0.2 | 5.4±0.3 | 5.2±0.2 | |
| **Ice flow speed, m / year** | Min | 547±103 | 149±31 | 113±58 | MEaSUREs2 (Rignot et al., 2017). |
| | Mean | 691±77 | 239±26 | 170±16 | |
| | Max | 758±55 | 378±12 | 188±7 | |





As both models necessitate the specification of bed slope, ice inflow speed, and glacier thickness prior to execution,
we conceptually represent the set of input parameters as (model, slope, thickness, speed), where 'model' denotes either
the viscous or viscoelastic formulation. Table 2 shows that for the considered glaciers, ice thickness ranges from
0.7±0.4 km to 2.4±0.2 km, bed slope varies from 0.01±0.01% to 5.4±0.3%, and ice flow speed can be as low
as113±58 m / year and as high as 758±55 m / year. Therefore, taking the bed slope in a range from 5% to 0.05%,
glacier thickness in a range from 1.0 to 2.5 km, and ice inflow speed in a range from 100 to 800 m/year, we ensure
that ~95% of the measurements align with the model setup (see Table S1).
Once the range of input parameters was established, we examined the sensitivity of the models to mesh size by running
them with the same set of parameters but varying mesh sizes at the lower domain surface (from 10 m to 250 m with
10 m step), while keeping the mesh size constant (250 m) at the upper surface of the glacier. Overall, to establish the
most efficient mesh size, 200 grounding zone width values were obtained and analyzed (Figure S1). The accuracy of
the viscoelastic model is more significantly affected by mesh size than the viscous model. For example, grounding
zone width values for glaciers with thicknesses of 2.5 km and 1 km, both with an inflow speed of 100 m/year, converge
to approximately 1.45 km for a mesh size of 250 m. However, at a mesh size of 10 m, these values were 0.96 km and
0.84 km, respectively (Figure S1 (d)). Comparing the dependences for the same slope of 5% for both models, we
conclude that for glaciers with the same thickness, lower ice flow speed is mere sensitive to the mesh size (red and
black dots in Figure S1 (c) and (d)).
We empirically determined that the average accuracy of manual mapping is approximately 200 m, therefore as long
as the model outputs do not deviate by more than 0.2 km from the asymptotic value of the grounding zone width, we
can conclude that the mesh impact lies withing the confidence interval of manual mapping. The noticeable accuracy
deterioration, exceeding 200 m, occurs at a mesh size of 210 m for the viscous model (Figure S1 (a)), and 200 m for
the viscoelastic model (Figure S1 (d)). However, for the viscoelastic model, we observe several step-like changes in
the grounding zone width value, with the first noticeable shift takes occurring at a mesh size of 60 m (Figure S1 (b)).
Therefore, to ensure the greatest possible modelling precision and maintain the consistency of the results, we have
chosen 50 m as the mesh size at the lover domain boundary for the following main analysis.
The main analysis of the grounding zone evolution depending on physical representation (viscous or viscoelastic) and
ice-bed system parameters was carried out retaining a constant mesh size of 50 m and 250 m at the lower and upper
domain boundaries, respectively, which was previously determined as the most efficient. Maintaining a consistent
glacier domain length of 20 km for all model runs, various parameter tests were conducted, encompassing ice
thicknesses of 1.0, 1.5, 2.0, and 2.5 km; horizontal ice inflow speeds of 100, 350, 600, and 800 m/year; and bedrock
slopes of 5.0, 4.5, 4.0, 3.5, 3.0, 2.5, 2.0, 1.5, 1.0, 0.5, 0.1, and 0.05% (see Figure S2). Therefore, a total of 192 sets of
initial parameters were investigated for each model, covering all possible combinations of the specified ice thickness,
ice inflow speed, and bedrock slope values. For each parameter set, both the viscous and viscoelastic models were
initially run for a duration of two months within the model's time frame, assuming a stationary ocean with no tides to
allow the model to reach stability. Subsequently, the models were run over a 7-day period with tides incorporated. In
the still water scenario, the water level corresponds to the low tide situation in the tidal problem. The choice of a one-
week time limit for the tidal problem allows the model to adapt to tidal impacts and enhances results accuracy. In most



tidal model runs, the grounding zone width slightly increases within the first 3 to 5 days with each tide while the
models adapt and stabilizes afterward. The resulting grounding zone width value for each model run is determined as
the average of the grounding zone width values for the last two days.
The source code of the viscous model, developed by (Stubblefield et al., 2021), was used as a basis of the viscoelastic
model (see Code and Data availability). Necessary adjustments to the mesh size and glacier parameters for both
publicly available source codes were made accordingly. Consequently, a total of 1,168 model runs were performed
while conducting the research: 400 runs for the mesh sensitivity analysis and 768 runs for the main analysis, which
includes the grounding zone width dependence analysis from the main glacier parameters for both models. As for the
grounding line generation two model runs are required, 584 grounding zone values were obtained: 200 for the mesh
sensitivity analysis and 384 for the main analysis. In total, these code runs required about 1400 hours (~58 days) of
continuous computations.
**5.    Discussion**
**5.1.    Modeled tide-induced grounding zone dependence from ice-bed system parameters**
A total of 384 grounding zone width values were generated utilizing all possible combinations of selected ice-bed
system parameters (Figure S2), while maintaining constant mesh sizes of 250 m and 50 m at the upper and lower
glacier surfaces, respectively. These grounding zones are illustrated in Figure 3, where they are grouped by the bedrock
slope for each model. Figure 3 shows the dependence of the grounding zone width on the glacier thickness for each
bed slope, with the outputs color-coded based on the inflow speed. The relationship between modeled grounding zone
width ($GZ$) and ice thickness ($H$) for each inflow speed and bed slope can be approximated by a linear function $GZ =$
$a \cdot H + b$, where coefficients $a$ and $b$ are unique for each model formulation, bed slope, and ice inflow speed. The
approximation equations with corresponding coefficients of determination ($R^2$ values) are provided in Table S2,
where $R^2$ ranges from 0.902 to 1.000 for the viscous model and from 0.874 to 1.000 for the viscoelastic model,
showing a high linearity of the grounding zone dependence on the glacier thickness for any bedrock slope if ice speed
is constant.
The linear approximation $GZ = a \cdot H + b$ of the values shown in Figure 3, performed separately based on the bed
slope for each inflow speed, facilitates tracking the evolution of the grounding zone dependence on ice flow speed as
the bed slope increases. Denoting $a_{100}, a_{350}, a_{600}, a_{800}$ as slope coefficients for ice inflow speeds of 100, 350, 600,
and 800 m/year, respectively, reveals the following evolution of their relative magnitudes: $a_{100} > a_{350} > a_{600} > a_{800}$
for bed slope $5.0\% \geq \alpha \geq 3.0\%$ and $\alpha = 0.5\%$, $a_{100} > a_{600} > a_{350} > a_{800}$ for bed slope $2.5\% \geq \alpha \geq 1.5\%$,
$a_{600} > a_{100} > a_{800} > a_{350}$ for bed slope $\alpha = 1.0\%$, $a_{350} > a_{100} > a_{600} > a_{800}$ for bed slope $\alpha = 0.1\%$, and
$a_{800} > a_{600} > a_{350} > a_{100}$ for bed slope $\alpha = 0.05\%$ for the viscous model (see Table S2 and Figure 3). Analogously,
for the viscoelastic model, the slope coefficients demonstrate the following pattern: $a_{100} > a_{350} \geq a_{600} > a_{800}$ for
bed slope $5.0\% \geq \alpha \geq 4.5\%$ and $2.5\% \geq \alpha \geq 2.0\%$, $a_{350} > a_{100} > a_{600} \geq a_{800}$ for bed slope $4.0\% \geq \alpha \geq 3.5\%$
and $\alpha = 1.5\%$, $a_{100} > a_{600} > a_{350} > a_{800}$ for bed slope $\alpha = 3.0\%$, $a_{100} > a_{350} > a_{800} > a_{600}$ for bed slope $\alpha =$
$1.0\%$, $a_{600} > a_{350} > a_{800} > a_{100}$ for bed slope $\alpha = 0.5\%$, $a_{800} > a_{600} > a_{350} > a_{100}$ for bed slope $0.1\% \geq \alpha \geq$
$0.05\%$. Therefore, at steeper slopes, both models exhibit the same ratio of slope coefficients, namely, $a_{100} > a_{600} >$

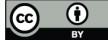



$a_{350} > a_{800}$, which changes to the reverse ratio at shallower slopes: $a_{800} > a_{600} > a_{350} > a_{100}$. This indicates that at steeper slopes, a grounding zone is more sensitive to changes in glacier thickness if the ice flow is slow, while for almost flat bedrocks, a grounding zone width is more affected by variations in ice thickness of faster-flowing glaciers.



**Figure 3.** Dependence of the grounding zone width from the glacier thickness for all considered inflow speeds and bed slopes for both viscous and viscoelastic models. Subplots (a) – (l) correspond to the viscous model; subplots (m) – (x) correspond to the viscoelastic model. Corresponding bed slope is written above each subplot, the x-axis of each subplot shows the glacier thickness in meters, while the y-axis shows the evolution of the grounding zone as the





glacier becomes thicker. Each subplot contains four sets of values, colored based on the inflow speed used as a model
input at a corresponding model run.
Additionally, Figure 3 and Table S2 also show that the linear dependence $GZ = a \cdot H + b$ becomes steeper as the bed
slope decreases, which, in terms the slope coefficient $a$, means that the coefficient's magnitude increases as the
bedrock becomes shallower. We conducted further analysis using the data from Figure 3 by subtracting the grounding
zone widths corresponding to the thickest and the thinnest glaciers ($\Delta GZ$) for each bed slope and ice flow speed. While
this analysis aimed to assess the impact of glacier thickness on the grounding zone for different bed slopes, Table S3
confirms the previous conclusion that the grounding zone at steep bed slopes is more sensitive to lower flow speeds,
as evidenced by the descending order of $\Delta GZ$ values in the column corresponding to 5.0% bed slopes for both models.
Conversely, both models exhibit an ascending order of $\Delta GZ$ values in 0.05% column, indicating a higher sensitivity
of grounding zone to ice thickness for faster flowing glaciers.
The $\Delta GZ$ values, averaged between those corresponding to different flow speeds for each bed slope and denoted as
'Mean' in Table S3, increase as the bed slope decreases. This pattern is observed for both models, with the mean
difference values being larger for the viscous model. At a 5.0% bed slope, the difference in mean $\Delta GZ$ does not exceed
10 meters: 98 m versus 105 m for the viscous and viscoelastic models, respectively. The difference in $\Delta GZ$ remains
similar between 5.0% and 1.0% bed slopes, while $\Delta GZ$ for the viscoelastic model is less than two times greater than
$\Delta GZ$ for the viscous model. However, at a 0.5% bed slope, $\Delta GZ$ for the viscous model becomes greater than $\Delta GZ$ for
the viscoelastic model. At a bed slope of 0.05%, $\Delta GZ$ for the viscous model is almost 2.4 times greater than that for
the viscoelastic model: 6130 m versus 2543 m for viscous and viscoelastic models, respectively. Moreover, the
viscous model predicts a ~62 times enlargement of $\Delta GZ$ if the bed slope changes from 5.0% to 0.05%, while the
viscoelastic model forecasts a ~24 times enlargement of $\Delta GZ$ for the same slope change.

### 5.2. Model validation with DInSAR grounding zone measurements

In addition to the grounding zone, along each of 80 profiles we calculated the average values of bedrock slope, glacier
thickness, and flow speed. While grounding zones were used to verify the models' performances, bed slopes, ice
thicknesses, and flow speeds were used as input parameters. Therefore, as every profile is characterized by three input
measurements, a total of 240 input measurements were performed. We test the models using bed slopes varying from
5% to 0.05% with ice thicknesses ranging from 1.0 to 2.5 km, and ice inflow speeds from 100 to 800 m/year. This
choice of model input parameters ranges ensures that ~97% of the input measurements, accounting for the
corresponding measurement errors, fall within the specified ranges (Table S1).
The relative distribution of input measurements is shown in Figure S3. All the ice flow measurements (Figure S3) fall
between 100 m/year and 800 m/year, with REN's speed measurements being smaller than 200 m/year, MU's values
ranging between 150 and 400 m/year, and TOT's speeds exceeding 500 m/year. REN does not have slopes lower than
0.8%, while MU and TOT have shallower slopes. Histogram (j) in Figure S3 shows high density of measurements
clustered between 0% and 0.2% bedrock slopes if not accounting for the measurement errors. Due to computational
limitations, we are unable to model bedrock slopes shallower than 0.05%. However, considering bed slopes associated
errors (Figure 4), the minimum bed slope of 0.05% ensures that all the measurements along shallow beds fall into the





modeled range of bed slopes (from 0.05% to 5%). Three bed slope measurements are greater than 5.0%, with two
belonging to MU (profiles 27 and 28 in Table S1) and one to REN (profile 0). The Interquartile Range (IQR) method
of outlier removal, which classifies a data point as an outlier if it exceeds the 25th percentile of the dataset by more
than $1.5 \cdot IQR$ or falls behind the 75th percentile by more than $1.5 \cdot IQR$, detected these three measurements as outliers
(empty dots in box plot E1 in Figure S3). Only four thickness measurements, all belonging to REN
(profiles 0, 1, 2, and 3), are less than 1 km (histograms (h) and (k) in Figure S3), and were classified by the IQR-based
method as outliers as well (empty dots in box plot (n) in Figure S3). These seven measurements (four for ice thickness
and three for bed slope), determined by the IQR-based method as outliers belong to six profiles: four for REN (profiles
0, 1, 2, and 3), and two for MU (profiles 27, 28). We assess the models' capabilities to model DInSAR-observed
grounding zones including and excluding these six profiles (Figure 4d).
Figure 4 provides the comparison of the viscous and viscoelastic models with each other and with the remote sensing
observations over MU, TOT, and REN glaciers. Figure 4 (a) and Figure 4 (b) present the grounding zone width
obtained for the viscous and viscoelastic models, respectively, where the results for different inflow speeds are
averaged by glacier thickness. Error bars in Figure 4 (a) and Figure 4 (b) represent critical grounding zone width
values, which are dependent on ice speed for a given ice thickness. The grounding zone width for both models
increases as the bedrock slope decreases. The steepest dependence is observed for the smallest tested glacier thickness
(1 km), while the shallowest dependence, resulting in the largest grounding zone width values, characterizes the
thickest glaciers (2.5 km). The grounding zone width values of the viscoelastic model ($GZ_{VE}$) plotted against the
viscous model's outputs $GZ_V$, as shown in Figure 4 (d), exhibit a linear relationship: $GZ_{VE} = 0.49 \cdot GZ_V + 0.47$, with
a coefficient of determination ($R^2$) of 0.97. Consequently, for any combination of bedrock slope, glacier thickness,
and ice inflow speed, the grounding zone width obtained from the viscoelastic model is nearly half that of the
grounding zone width calculated by the viscous model on shorter time scales.
As each profile is characterized by a specific slope, thickness, and speed measurement, the measurements falling
outside the chosen ranges pertain to six profiles: four for REN (profiles 0, 1, 2, and 3), and two for MU (profiles 27
and 28). These profiles are labeled as 'extra' profiles in Table S2. We performed the assessment of the models'
capabilities to replicate DInSAR-observed grounding zones, both including and excluding these 'extra' profiles.
Figure 4 C indicates the superimposed outputs of the viscous and viscoelastic models alongside the DInSAR
grounding zone measurements overlaid on the modeling results, where empty circular markers correspond to the
grounding zones extracted along the 'extra' profiles.
Considering all 80 profiles and disregarding the measurements' error bars, ~41%, ~0%, and ~14% of TOT's, REN's,
and MU's measurements, respectively, fall into the viscous model's domain. Meanwhile, ~88% of TOT's
measurements, 100% of REN's measurements, and ~71% of MU's measurements, all without considering the error
bars, are successfully accommodated by the viscoelastic model. When including the error bars in consideration, the
performance of the models significantly improves. For the viscous model, the percentage of successfully modeled
measurements increases from ~41% to ~65% for TOT, from ~0% to ~26% for REN, and from ~14% to ~39% for
REN. For the viscoelastic model, this performance improvement is evident in the following notable expansions: from
~88% to ~100% for TOT, from ~71% to ~84% for MU and remains consistently at 100% for REN. Excluding the



'extra' profiles, ~29% versus ~57% of the TOT's measurements fall within the domain of the viscous model when
disregarding and considering the measurements' error bars, respectively. Analogously, for the viscous model, ~0%
turns into ~33% for REN, and ~13% transforms into ~47% for MU when taking the measurements' error bars into
account. Disregarding the 'extra' profiles, the difference in the viscoelastic model's performance, when ignoring and
considering the error bars, changes from ~86% to ~100% for TOT, increases from ~76% to ~87% for MU, and
remains unchanged at 100% for REN.
Determining model's accuracy as the percentage of DInSAR measurements that fall inside the domain of a
corresponding model, for the viscoelastic model, REN consistently demonstrates 100% accuracy regardless of whether
the 'extra' profiles are considered, and whether error bars are included or not. For TOT, the accuracy remains at 100%
when the error bars are included, with or without the 'extra' profiles, while without the measurements' error bars, the
'extra' profiles improve accuracy by only ~2%. For MU, the inclusion of the 'extra' profiles results in a ~5% accuracy
increase without the error bars, and a ~3% increase with the error bars. Conversely, for the viscous model, the
inclusion of the 'extra' profiles improves the accuracy only for TOT: without the 'extra' profiles, the accuracy improves
by ~8% and ~12% with and without the error bars, respectively. However, the accuracy of the viscous model
decreases by ~6% and 8~% for REN and TOT, respectively, when the 'extra' profiles are removed and the
measurements' error bars are considered. Thus, discarding the 'extra' profiles does not significantly enhance the
models' performances and may even reduce the percentage of successfully modeled measurements in some cases.
Overall, considering all three glaciers together and accounting for all 80 profiles, the viscous model achieves ~16%
or ~41% accuracy without or with the measurements error bars, respectively, while the viscoelastic model achieves
~81% or ~91% accuracy without or with the measurements error bars, respectively. Excluding the 'extra' profiles, the
accuracy of the viscous model improves from ~13% to ~46% when the measurements' error bars are considered,
while the accuracy of the viscoelastic model changes from ~84% to ~93% when the measurements' error bars are
taken into account. Therefore, excluding the 'extra' profiles and considering error bars, the viscoelastic model
outperforms the viscous model by ~47%. However, without error bars, the viscoelastic model outperforms the viscous
model by ~71%. This finding underscores the critical importance of incorporating the elastic component in Stokes-
based fluid glacier formulations.



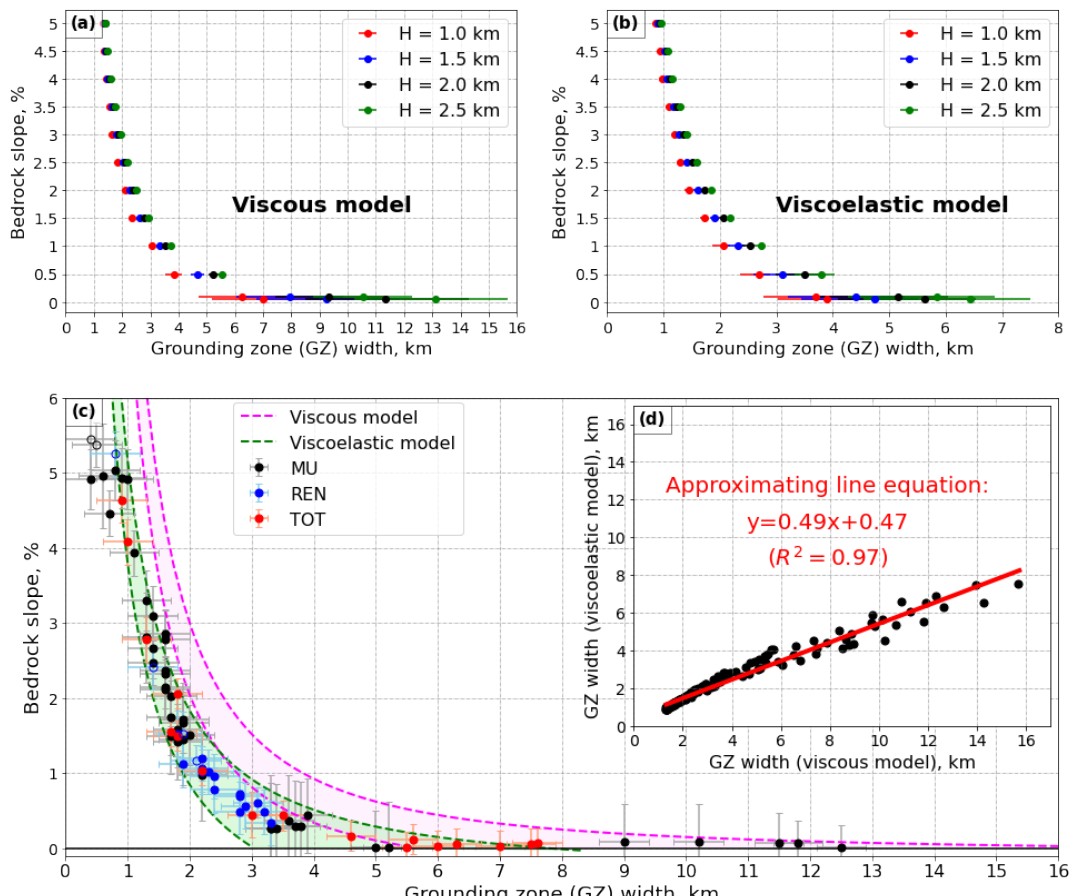


**Figure 4.** Modeling results of (a) viscous and (b) viscoelastic models, averaged by the thickness values, with error
bars, representing the distance between the averaged grounding zone width for given thickness and
maximum/minimum grounding zone (GZ) values for this thickness but different inflow speed values. (c) comparison
of the modelling results (pink and green areas) with the DInSAR grounding zone measurements, where empty markers
show grounding zones obtained along 'extra' profiles from Table S2; (d) correlation plot of the modelling results.

## 6. Conclusion

The variational formulation described in this paper has been successfully applied to the modeling of viscous and
viscoelastic glaciers on prograde bedrock slopes. In the investigated models, a thin water layer propagates into the gap
between the lower glacier surface and the bedrock, which forms due the impact of tidal forces. This gap formation
leads to the inland movement of the grounding line, resulting in the formation of a grounding zone, which represents
the magnitude of tide-induced grounding line migration over one tidal cycle. To compute the grounding zone width,
the viscoelastic and viscous models both require bed slope, glacier thickness, and glacier flow speed as input
parameters. Considering the grounding zone dependence on these parameters, several conclusions have emerged.



Firstly, the grounding zone widens as the bed slope becomes shallower, which is consistent with previous
observational studies (Milillo et al., 2017, 2019, 2022; Chen et al., 2023b). Secondly, for a given bed slope, the
grounding zone is wider for a thicker glacier, which is not only evident from the modeling results but is also supported
by the DInSAR measurements (Figure 5 a and b). Considering DInSAR measurements characterized by bed slopes
ranging from 0.1% to 0.5% and ice flow speeds ranging from 100 to 350 m/year and overlaying them with modelling
results obtained for the same range of bed slopes and flow speeds, we obtain a linear correlation between glacier
thickness and grounding zone width (Figure 5 a). Analogously, in Figure 5 b, a similar linear relationship between
glacier thickness and grounding zone is observed for the same range of glacier flow speeds and bed slopes ranging
from 1.6% to 2.5%. The range of slopes was increased compared to Figure 5 a as the grounding zone sensitivity to
variations in bed slopes decreases when bed becomes steeper, according to the first conclusion we made. Lastly, it can
be concluded from the modeled grounding zones that on steep bed slopes, the grounding zone is more responsive to
changes in glacier thickness when ice flow is slow; on shallow (mostly flat) bedrocks, the grounding zone is more
sensitive to variations in ice thickness if a glacier is flowing rapidly. Confirmation of this modelling result over shallow
slopes using DInSAR data is shown in Figure 5 c, where measurements characterized by faster ice flow exhibit a
steeper dependence of grounding zone width on glacier thickness compared to slower glacier flow. However, due to
the sparseness of the DInSAR dataset at steeper slopes, we cannot confirm the modeling-derived conclusion for steep
sloped bedrock.

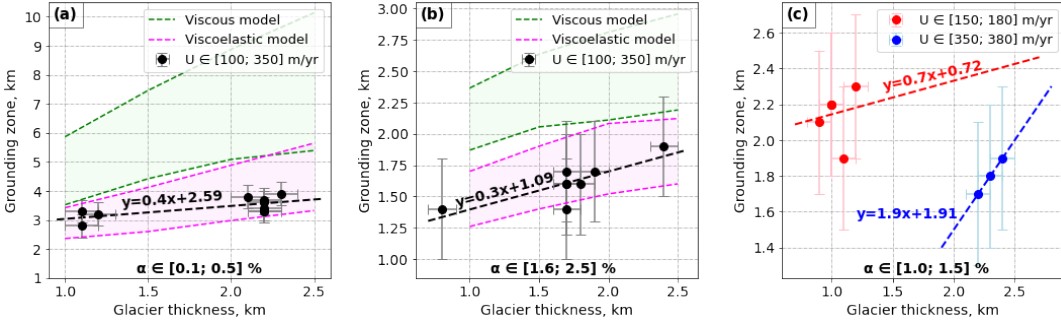


**Figure 5.** Relationship between glacier thickness and DInSAR-derived grounding zone width. (a) DInSAR
measurements of grounding zone width (black dots) conducted along profiles characterized by bed slopes ranging
from 0.1 to 0.5% and ice flow speeds from 100 to 350 m/year. The black dashed line describes shows the linear
correlation between the ice thickness and DInSAR-derived grounding zones. The green and pink areas represent the
grounding zones calculated by the viscous and viscoelastic models, respectively, using the same range of bed slopes
and flow speeds as input parameters. (b) The black dots and the dashed line correspond to DInSAR measurements
described by bed slopes ranging from 1.6 to 2.5% and ice flow speeds from 100 to 350 m/year, and their linear
approximation, respectively. The green and pink areas correspond to the grounding zones, calculated by the viscous
and viscoelastic models, respectively, for the same range of bed slopes and flow speeds. (c) Grounding zone
measurements over 1.0 to 1.5% bed slopes, described by [150; 180] m/year and [350; 380] m/year ice flow speeds
with corresponding linear approximations.



Comparing all grounding zone values generated by the viscous and viscoelastic models, we observe a linear
relationship between these values, with the grounding zone width obtained from the viscous model being
approximately twice that calculated by the viscoelastic model across varying bedrock slopes, glacier thicknesses, and
ice inflow speeds. To validate the models' performance, we compare their grounding zone outputs with DInSAR-
derived grounding zones over Moscow University, Totten, and Rennick glaciers. To ensure the fair comparison of the
models' outputs and the measurements, we input identical bed slope, glacier thickness, and glacier flow speed as those
corresponding to these three glaciers and account for respective errors.
Comparison of the grounding zones, obtained from the DInSAR interferograms, with the modeled grounding zones
shows that the viscoelastic model achieves significantly higher accuracy than the viscous model. Accounting for the
error bars of the DInSAR measurements, the viscoelastic model successfully reproduced ∼93% of all the
measurements, while the viscous model succeeds with ∼46% of the measurements. If the error bars are not considered,
∼84% versus ∼13% of the measurements are replicated by the viscoelastic and viscous model, respectively. Therefore,
the accuracy of the viscoelastic model outperforms the accuracy of the viscous model by up to ∼71%. Notably, the
viscoelastic model reproduces all the measurements over Rennick glacier, either with or without the error bars, while
the viscous model fails to replicate a single Rennick grounding zone measurement when the error bars are not included.
These observations highlight the significance of incorporating the elastic component in Stokes-based glacier modeling
compared to a purely viscous model.
Significant difference between viscous and viscoelastic models can be explained from a continuum mechanics
perspective. Viscous response to deformation occurs over long timescales and corresponds to gradual deformations.
However, a tidal impact occurs within a single day, rendering tide-induced deformations too rapid for accurate
representation by a purely viscous model. Therefore, an element responsible for rapid deformations, or an elastic
component, becomes necessary. Putting the viscous and elastic components in series, known as the Maxwell model
of viscoelasticity, we ensure that both slow and rapid deformations are taken into account. However, the simple
Maxwell model describes small deformations, whereas the deformations of our interest may extend up to 50% of the
glacier domain length. Therefore, we applied the upper-convected Maxwell model of viscoelasticity, which includes
some geometrical non-linearity and allows the modelling of significantly larger deformations compared to the simple
Maxwell model.
Finally, (Li et al., 2023) mention that both ICESat laser altimetry and Sentinel-1a/b three-image DInSAR
interferometry failed to delineate main trunk of TOT glacier and the central part of the MU main trunk due to the fast
ice flow in these regions. On the contrary, the four-image CSK DInSAR technique utilized in this study allowed us to
map grounding lines even over these fast-flowing areas. (Li et al., 2023) estimated the average grounding line retreat
between 1996 and 2020 as $3.51 \pm 0.49$ km for the southern lobe of the TOT main trunk, and as 13.85 km and 9.37 km
for the western and eastern flanks of the MU main trunk, respectively. As, according to (Li et al., 2023), it is impossible
to determine the magnitude of tidally induced grounding line migrations in 1996 from the historic grounding line
dataset (Rignot et al., 2016), we assume the 1996 grounding line position as the average position between high and
low tides. To calculate the long-term retreat, we estimate the distance from the historic grounding line to the center of
the DInSAR-derived grounding zones for each glacier of interest. As a result, for MU, between 1996 and 2021, we



detect an average retreat of the main trunk of 9 ± 2 km, with 18 ± 1 km retreat at the western flank, 6.7 ± 0.6 km retreat at the central part of the main trunk, and 4.2 ± 0.6 km retreat at the eastern flunk. Therefore, the western flank demonstrates the highest retreat rate of 690 ± 40 m/year, while the average glacier retreat rate over this period was 340 ± 80 m/year. For TOT, between 1996 and 2020, we observe an average retreat of the main trunk of 9 ± 3 km with 13.9 ± 0.1 km retreat at the western flank, 17 ± 1 km retreat at the central part of the main trunk, and 5.2 ± 0.3 km retreat at the eastern flank. Therefore, while the average rate of TOT retreat between 1996 and 2020 was 360 ± 120 m/year, the central part of the main trunk retreated as fast as 680 ± 40 m/year. In the meantime, the position of the REN grounding line at the main trunk did not change between 2000 and 2020, which signifies the stability of the glacier over the past 20 years.

**Appendix A: Glacier modelling**

Here, we provide a detailed description of the viscoelastic model and compare it with the viscous model.

**A1. Principal notation**

Notation, used in the paper, is listed in Table A1.

**Table A1.** Models' principal notation

| Symbol | Quantity |
|---|---|
| \multicolumn{2}{c}{Geometry-related quantities} | |
| $(X, Y)$ | laboratory coordinate system |
| $x = (x, y)$ | spatial point with coordinates $x$ and $y$ |
| $\Omega$ | glacier domain |
| $\partial\Omega$ | boundary of the glacier domain |
| $\Gamma_D$ | inflow boundary |
| $\Gamma_N$ | outflow boundary |
| $\Gamma_a$ | ice–air surface |
| $\Gamma_b$ | ice–bedrock surface |
| $\Gamma_w$ | ice–water surface |
| $\Gamma_s$ | lower glacier boundary |
| $h(x, t)$ | surface elevation of the ice shelf |
| $s(x, t)$ | lower boundary of the ice shelf |
| $\alpha$ | bedrock slope |
| $b(x)$ | bedrock slope function |
| $A$ | bedrock inclination parameter |
| $L$ | glacier length |
| $H$ | glacier thickness |
| $l$ | sea level |
| \multicolumn{2}{c}{Materials properties} | |
| $\rho_i$ | ice density |
| $\rho_w$ | water density |
| $\phi$ | friction |
| $C$ | friction coefficient |
| $\eta$ | ice viscosity |
| $G$ | sheer modulus |



| | | |
|---|---|---|
| $\lambda$ | relaxation time | |
| $r$ | stress exponent (from the Glen's flow law) | |
| $A$ | ice softness | |
| **Scalar quantities** | | |
| $t$ | time | |
| $p$ | ice pressure | |
| $p_w$ | water pressure at the ice–water interface | |
| $p_w^0$ | hydrostatic water pressure | |
| $v_0$ | inflow speed on $\Gamma_D$ | |
| **Vector quantities** | | |
| $\hat{n}(x)$ | unit outward normal vector at point $x$ of a domain boundary | |
| $v$ | ice flow velocity | |
| $b$ | body force | |
| $g$ | acceleration due to gravity | |
| **Tensor quantities** | | |
| $\mathbb{I}$ | identity tensor | |
| $\mathbb{P}$ | orthogonal projection onto the boundary | |
| $D$ | strain rate tensor | |
| $T$ | Cauchy stress tensor | |
| $\tau$ | deviatoric stress tensor | |
| **Mathematical operators** | | |
| $\nabla$ | spatial gradient operator | |
| $\nabla \cdot$ | spatial divergence operator | |
| $\cdot$ | inner (dot) product | |
| $\otimes$ | tensor product | |
| $\overset{\nabla}{\tau}$ | upper-convected time derivative of some tensor field (in this case, of the tensor $\tau$) | |
| **Model parameters** | | |
| $\delta \ll 1$ | Glen's flow law parameter numerical parameter, used to prevent model numerical instabilities | |
| $B$ | Glen's flow law parameter | |

**A2. Model formulation**

Using the notation provided in Table A1, we describe the formulation of both models, which have the same domain
(see Viscous and viscoelastic models) and boundary conditions, but different governing equations.

**A2.1.   Governing equations**

In both models, a glacier behaves as an incompressible non-Newtonian fluid, either viscous or viscoelastic.
Incompressibility implies that the fluid density does not change during flow, which is mathematically infers zero
divergence of the flow velocity $v$:

$$\nabla \cdot v = 0. \tag{A1}$$

Both models are described by the Cauchy's first law of motion under quasi-static conditions, which provides the
momentum conservation, and is identified as:

$$\nabla \cdot T(v, p) + \rho_i g = 0, \tag{A2}$$

where $T$ is the Cauchy stress tensor, $\rho_i$ is the ice density, and $g$ is vector of gravitational acceleration, which in the
glacier reference system is identified as $g = g(0 \quad -1)^T$ with magnitude $g$.



The difference between the models becomes apparent when considering the constitutive law, defining the physical nature of the models. The viscous model is described by the following equation:

$$\boldsymbol{T}(\boldsymbol{v}, p) = -p\mathbb{I} + 2\eta(\boldsymbol{v})\boldsymbol{D}, \tag{A3}$$

where $p$ is the ice pressure, $\mathbb{I}$ is a second-order identity tensor, $\eta(\boldsymbol{v})$ is a velocity-dependent ice viscosity, and $\boldsymbol{D}$ is a strain rate tensor

$$\boldsymbol{D}(\boldsymbol{v}) = \frac{1}{2}[\nabla\boldsymbol{v} + (\nabla\boldsymbol{v})^T]. \tag{A4}$$

Ice viscosity in the viscous model is identified via Glen's flow as

$$\eta(\boldsymbol{v}) = 2^{\frac{-1-n}{2n}} \cdot \sqrt[-n]{A}(|\boldsymbol{D}(\boldsymbol{v})|^2 + \delta)^{\frac{1-n}{2n}}, \tag{A5}$$

where $n \geq 1$ is the stress exponent, $A > 0$ is the ice softness, and $\delta \ll 1$ is an infinitesimal numerical parameter, used to prevent numerical instability of the models at zero strain rate.

For the viscoelastic model, constitutive law (A3) and the viscosity expression (A5) are principally different. We consider the Maxwell model of viscoelasticity, which considers both viscous and elastic components assuming that deformation properties can be represented by a purely elastic spring and a purely viscous dashpot connected in series. Therefore, in the Maxwell model, a viscoelastic material behaves as a purely viscous flow under slow deformation (long timescale), while it exhibits elastic resistance to rapid deformations (short timescale). However, as the simple Maxwell model describes small deformations, we, apply the upper-convected Maxwell model instead, which includes some geometrical non-linearity. The constitutive relation for the viscoelastic model is identified as

$$\boldsymbol{T}(\boldsymbol{v}, p) = -p\mathbb{I} + \boldsymbol{\tau}, \tag{A6}$$

where, compared to the purely viscous model (A3), the strain rate tensor $\boldsymbol{D}$ (equation (A4)) is replaced with the deviatoric stress tensor $\boldsymbol{\tau}$, which is strain rate-dependent:

$$\boldsymbol{\tau} + \lambda\overset{\triangledown}{\boldsymbol{\tau}} - 2\eta(\boldsymbol{\tau})\boldsymbol{D}(\boldsymbol{v}) = 0, \tag{A7}$$

where $\lambda = \frac{\eta(\tau)}{G}$ is the relaxation time with the sheer modulus $G$, and the ice viscosity is

$$\eta(\boldsymbol{\tau}) = \frac{1}{2A|\boldsymbol{\tau}|^{n-1}}. \tag{A8}$$

$\overset{\triangledown}{\boldsymbol{\tau}}$ represents the upper-convected time derivative of $\boldsymbol{\tau}$:

$$\overset{\triangledown}{\boldsymbol{\tau}} = \frac{D\boldsymbol{\tau}}{Dt} - (\nabla\boldsymbol{v})^T \cdot \boldsymbol{\tau} - \boldsymbol{\tau} \cdot \nabla\boldsymbol{v}, \tag{A9}$$

where $\frac{D\boldsymbol{\tau}}{Dt} = \frac{\partial\boldsymbol{\tau}}{\partial t} + v\nabla\boldsymbol{\tau}$ is the material derivative of $\boldsymbol{\tau}$. Partial time derivative of $\boldsymbol{\tau}$ on the current time step is calculated in the model through the previous time step using backward Euler approximation:

$$\frac{\partial\boldsymbol{\tau}(x, t)}{\partial t} = \frac{\boldsymbol{\tau}(x, t) - \boldsymbol{\tau}(x, t - \varDelta t)}{\varDelta t}. \tag{A10}$$

**A2.2.   Evolution of the lower boundary**

The time evolution of the lower boundary $y = s(x, t)$ is governed by the kinematic equation, which expresses the fact that the surface moves with the ice flow, and under the assumption that there are no mass changes at the lower surface, such as melting or freezing, can be written as (Hirt and Nichols, 1981; Schoof, 2011)



$$\frac{\partial s}{\partial t} + v_x \frac{\partial s}{\partial t} = v_y, \tag{A11}$$

where $v_x$ and $v_y$ are the components of the surface velocity vector $\boldsymbol{v}|_s = \left(v_x, v_y\right)^T$. Rewriting equation (A11) in terms

of the outward-pointing normal to the lower boundary $\widehat{\boldsymbol{n}}|_s = \frac{\left(\frac{\partial s}{\partial x} \quad -1\right)^T}{\sqrt{1+\left(\frac{\partial s}{\partial x}\right)^2}}$, we get

$$\frac{\partial s}{\partial t} = -\boldsymbol{v} \cdot \widehat{\boldsymbol{n}}|_s \cdot \sqrt{1 + \left(\frac{\partial s}{\partial x}\right)^2}. \tag{A12}$$

As the solution of equation (A12) is numerically unstable (Durand et al., 2009b), we apply the backward Euler method
to get rid of the instability. Denoting the approximate solution on $k$-th time step as $s_*$, such as $s_* \equiv s(x, t_k)$, and
applying the backward Euler method to equation (A12) under the assumption that $\left(\frac{\partial s}{\partial x}\right)^2 \ll 1$, we get

$$s_*(x, t_k) = s(x, t_k - \Delta t) - \Delta t \cdot v_n(x, s, t_k), \tag{A13}$$

where $\boldsymbol{v} \cdot \widehat{\boldsymbol{n}}|_s$ was replaced with $v_n$. We assume that the ocean is hydrostatic and define $p_w$ as the water pressure at
the ice–water interface and $p_w^0$ as the hydrostatic water pressure. If $l$ is sea level, the hydrostatic water pressure at $k$-
th time step is governed by the following equation:

$$p_w^0(x, s, t_k) = p_w g(l(t_k) - s_*(x, t_k)). \tag{A14}$$

### A2.3.   Boundary conditions

Identifying $\widehat{\boldsymbol{n}}(\boldsymbol{x})$ as a unit outward normal vector at some point $\boldsymbol{x}$ of any domain boundary, we determine an orthogonal
projection onto that boundary as a second-order tensor $\mathbb{P}$:

$$\mathbb{P} := \mathbb{I} - \widehat{\boldsymbol{n}}(\boldsymbol{x}) \otimes \widehat{\boldsymbol{n}}(\boldsymbol{x}), \tag{A15}$$

where $\otimes$ is the tensor product. Denoting $\cdot$ as an inner product, we also define the projection of the Cauchy stress tensor
$\boldsymbol{T}$ as

$$T_n = -\widehat{\boldsymbol{n}} \cdot \boldsymbol{T} \cdot \widehat{\boldsymbol{n}}. \tag{A16}$$

Both models use the same Dirichlet boundary conditions provided in Table A2, where $v_0 > 0$ is the horizontal ice
flow speed on the inflow boundary $\Gamma_D$.

**Table A2.** Models' boundary conditions

| Boundary | Boundary condition | Physical meaning of a boundary condition | Equation number |
|---|---|---|---|
| $\Gamma_w$ | $\boldsymbol{T} \cdot \widehat{\boldsymbol{n}} = -p_w \widehat{\boldsymbol{n}}$ | Stress continuity at the ice–water boundary | (A17) |
| $\Gamma_a$ | $\boldsymbol{T} \cdot \widehat{\boldsymbol{n}} = 0$ | No stress at the ice–air boundary | (A18) |
| $\Gamma_D$ | $\begin{cases} v_x = v_0 \\ \mathbb{P}\boldsymbol{T}\widehat{\boldsymbol{n}} = 0 \end{cases}$ | On the inflow boundary the horizontal velocity is uniform and there is no vertical shear stress | (A19) |
| $\Gamma_N$ | $\boldsymbol{T} \cdot \widehat{\boldsymbol{n}} = -\rho_i g(h - y)\widehat{\boldsymbol{n}}$ | Cryostatic normal-stress condition on the outflow boundary | (A20) |
| $\Gamma_b$ | $\mathbb{P}\boldsymbol{T}\widehat{\boldsymbol{n}} + \phi(\boldsymbol{v})\mathbb{P}\boldsymbol{v} = 0^*$ | Sliding law on the ice-bed boundary | (A21) |
| $\Gamma_b$ | $\begin{cases} T_n \geq p_w \\ v_n \leq 0 \\ (T_n - p_w)v_n = 0 \end{cases}$ | There are three possibilities for the normal stress and the normal velocity component on the ice-bed boundary: | (A22) |



| | | (1) The normal stress exceeds the water pressure ($T_n > p_w$) and the ice is not lifted off of the bed ($v_n = 0$); (2) The normal stress equals the water pressure ($T_n = p_w$) and the ice is lifted from the bed ($v_n < 0$); (3) The normal stress equals the water pressure ($T_n = p_w$), but the ice is not lifted from the bed ($v_n = 0$). | |
|---|---|---|---|
| | | * in equation (A21), $\phi(\boldsymbol{v}) = C(|\mathbb{P}\boldsymbol{v}|^2 + \delta)^{\frac{1-n}{2n}}$ is friction with friction coefficient $C$ | | |

**A3. Weak formulation**

In this subsection we provide the derivation of the viscoelastic model, while the viscous model is derived in (Stubblefield et al., 2021).

**A3.1.  Mixed formulation**

Let us define $V$ as the velocity function space. $K$ is a closed, convex subset of $V$ such that

$$K = \left\{\boldsymbol{v} \in V: \quad v_n|_{\Gamma_b} \leq 0 \quad and \quad v_x|_{\Gamma_D} = v_0\right\} \tag{A23}$$

Multiplying equation (A2) by $\boldsymbol{v} - \boldsymbol{u}$ (where $\boldsymbol{u} \in K$ is an arbitrary test function), and integrating the expression over the glacier domain $\Omega$, in the indicial notation we will get:

$$\int_\Omega (v_k - u_k)T_{kj,j}dV + \int_\Omega \rho_i g_k(v_k - u_k)dV = 0. \tag{A24}$$

Integrating the first integral in equation (A24) by parts and applying the divergence theorem (Green's identity), we then apply equation (A6) to one of the integrals and rewrite the resulting expression in tensor notation:

$$-\int_{\partial\Omega} \boldsymbol{T} \cdot \hat{\boldsymbol{n}} \cdot (\boldsymbol{v} - \boldsymbol{u})da + \int_\Omega \{-p\nabla \cdot (\boldsymbol{v} - \boldsymbol{u}) + \boldsymbol{\tau} \cdot \nabla(\boldsymbol{v} - \boldsymbol{u}) - \rho_i g(\boldsymbol{v} - \boldsymbol{u})\}dV = 0. \tag{A25}$$

Now we decompose $\partial\Omega$ onto the compounding boundaries (see equation (2)) and consider the first integral in equation (A25) over each boundary separately. Using equations (A15), (A16), and boundary condition (A21) on $\Gamma_b$, after integration over $\Gamma_b$ and taking into account that $T_n \geq p_w$ on $\Gamma_b$, we derive that

$$-\int_{\Gamma_b} \boldsymbol{T} \cdot \hat{\boldsymbol{n}} \cdot (\boldsymbol{v} - \boldsymbol{u})da \leq \int_{\Gamma_b} \alpha(\boldsymbol{v}) \cdot \mathbb{P}v \cdot \mathbb{P}(\boldsymbol{v} - \boldsymbol{u})da + \int_{\Gamma_b} p_w(v_n - u_n)da. \tag{A26}$$

On $\Gamma_w$, from equation (A17), we obtain the following expression:

$$-\int_{\Gamma_w} \boldsymbol{T} \cdot \hat{\boldsymbol{n}} \cdot (\boldsymbol{v} - \boldsymbol{u})da = \int_{\Gamma_w} p_w(v_n - u_n)da. \tag{A27}$$

On $\Gamma_D$, from equation (A19), we have $\mathbb{P}\boldsymbol{T}\hat{\boldsymbol{n}} = 0$, thus, this boundary does not contribute to the integral $\int_{\partial\Omega} \boldsymbol{T} \cdot \hat{\boldsymbol{n}} \cdot (\boldsymbol{v} - \boldsymbol{u})da$. On $\Gamma_a$, from equation (A18), we have $\boldsymbol{T} \cdot \hat{\boldsymbol{n}} = 0$, thus, this the ice-air boundary does not contribute to the integral $\int_{\partial\Omega} \boldsymbol{T} \cdot \hat{\boldsymbol{n}} \cdot (\boldsymbol{v} - \boldsymbol{u})da$ as well. On $\Gamma_N$, where from equation (A20), we know $\boldsymbol{T} \cdot \hat{\boldsymbol{n}} = -\rho_i g(h - y)\hat{\boldsymbol{n}}$, which means that the contribution from the boundary to $\int_{\partial\Omega} \boldsymbol{T} \cdot \hat{\boldsymbol{n}} \cdot (\boldsymbol{v} - \boldsymbol{u})da$ will be

$$-\int_{\Gamma_N} \boldsymbol{T} \cdot \hat{\boldsymbol{n}} \cdot (\boldsymbol{v} - \boldsymbol{u})da = \int_{\Gamma_N} \rho_i g(h - y)(v_n - u_n)da. \tag{A28}$$



Substituting equations (A26) – (A28) to equation (A25), replacing the union of $\Gamma_w$ and $\Gamma_b$ with $\Gamma_s$ and replacing $p_w$
with $p_w = \rho_w g(l - s + \Delta t \cdot v_n)$, which was derived from equations (A13) and (A14), we obtain

$$\int_\Omega \{-p\nabla \cdot (\boldsymbol{v} - \boldsymbol{u}) + \boldsymbol{\tau} \cdot \nabla(\boldsymbol{v} - \boldsymbol{u}) - \rho_i g(\boldsymbol{v} - \boldsymbol{u})\}dV + \int_{\Gamma_N} \rho_i g(h - y)(v_n - u_n)da +$$
$$+ \int_{\Gamma_b} \alpha(\boldsymbol{v}) \cdot \mathbb{P}\boldsymbol{v} \cdot \mathbb{P}(\boldsymbol{v} - \boldsymbol{u})da + \int_{\Gamma_s} \rho_w g(l - s + \Delta t \cdot v_n)(v_n - u_n)da \geq 0. \tag{A29}$$

We define Q as a function space for pressure ($q \in Q$), and $M$ as a function space for stress ($\mu \in M$). To shorten and
simplify the notation, we introduce following functions:

$$F(\boldsymbol{\tau}, \boldsymbol{v}, \boldsymbol{u}) = \int_\Omega \boldsymbol{\tau} \cdot \nabla \boldsymbol{u} - \rho_i g\boldsymbol{u}dV + \int_{\Gamma_N} \rho_i g(h - y)u_nda + \int_{\Gamma_b} \alpha(\boldsymbol{v}) \cdot \mathbb{P}\boldsymbol{v} \cdot \mathbb{P}\boldsymbol{u}da \tag{A30}$$

$$P(\boldsymbol{v}, \boldsymbol{u}) = \int_{\Gamma_s} \rho_w g(l - s + \Delta t \cdot v_n)u_nda \tag{A31}$$

$$d_\Omega(\boldsymbol{\mu}, \boldsymbol{\tau}, \boldsymbol{v}) = \int_\Omega \boldsymbol{\mu}\left(\tau + \frac{\eta}{G}\overset{\triangledown}{\tau} - 2\eta D(\boldsymbol{v})\right)dV \tag{A32}$$

$$b_\Omega(q, \boldsymbol{v}) = \int_\Omega q\nabla \cdot \boldsymbol{v}dV \tag{A33}$$

Writing inequality (A29) in terms of equations (A30) – (A33), we obtain

$$\begin{cases} F(\boldsymbol{\tau}, \boldsymbol{v}, \boldsymbol{v} - \boldsymbol{u}) + P(\boldsymbol{v}, \boldsymbol{v} - \boldsymbol{u}) - b_\Omega(p, \boldsymbol{v} - \boldsymbol{u}) \geq 0 \\ d_\Omega(\boldsymbol{\mu}, \boldsymbol{\tau}, \boldsymbol{v}) = 0 \\ b_\Omega(q, \boldsymbol{v}) = 0 \end{cases}. \tag{A34}$$

By analogy with (Stubblefield et al., 2021), we replace the mixed formulation (A34) with a penalty formulation

$$\begin{cases} F(\boldsymbol{\tau}, \boldsymbol{v}, \boldsymbol{u}) + P(\boldsymbol{v}, \boldsymbol{u}) - b_\Omega(p, \boldsymbol{u}) + \dfrac{\Pi'(\boldsymbol{v}, \boldsymbol{u})}{\epsilon} = 0 \\ d_\Omega(\boldsymbol{\mu}, \boldsymbol{\tau}, \boldsymbol{v}) = 0 \\ b_\Omega(q, \boldsymbol{v}) = 0 \end{cases}. \tag{A35}$$

Therefore, the penalized problem for the viscoelastic model is to find $(\boldsymbol{v}, p, \boldsymbol{\tau}) \in V \times Q \times M$, which satisfies the
boundary conditions and the system (A35).

**Code and Data availability**

All data needed to evaluate the conclusions in the paper are present in the paper and/or the Supplementary Materials.
We thank the Italian Space Agency (ASI) for providing CSK data (original COSMO-SkyMed product ASI, Agenzia
Spaziale Italiana (2008–2023)). Velocity (https://nsidc.org/data/NSIDC-0484/versions/2) and BedMachine
(https://nsidc.org/data/NSIDC-0756/versions/2) data products are available as MEaSUREs products at the National
Snow and Ice Data Center, Boulder CO (NSIDC). The source codes for the viscous and viscoelastic models are freely
available on https://github.com/agstub/grounding-line-methods/tree/v1.0.0 and
https://github.com/agstub/viscoelastic-glines GitHub repositories, respectively. Geocoded interferograms and
grounding-line positions are available at https://doi.org/10.5281/zenodo.10853336.



**Author contribution**

PM and NM designed the study; AS developed the viscous and viscoelastic models; NM performed the codes modifications and grounding zone simulations under the supervision of KN and RB; NM and PM performed the measurement of the grounding zones from the DInSAR data and the assessment of the main ice-bed system parameters; LD provided the CSK DInSAR data; NM and PN wrote the manuscript draft with contributions from KN, RB and AS reviewed and edited the manuscript.

**Competing interests**

The authors declare that they have no conflict of interest.

**Acknowledgements**

The research was conducted at the University of Houston, Houston, TX, US. We acknowledge the Research Computing Data Core (RCDC) for giving access to advance high-performance computing resources of the University of Houston. We thank the Italian Space Agency (ASI) for providing CSK data (original COSMO-SkyMed product ASI, Agenzia Spaziale Italiana (2008–2023)).

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
