# Peer review of "Importance of ice elasticity in simulating tide-induced grounding"

_EGUsphere, 2024_

## Author Comment (AC2)

**Response to Reviewer 1**

**Blue font:** comments from Reviewer 1
**Black bold font:** Authors' responses to the Reviewer's comments

**Note 1:** The Reviewers recommended major revisions, including restructuring the text and redoing certain measurements. The paper is almost entirely rewritten and highlighting these changes would significantly hinder the manuscript's readability. Therefore, the revised version does not highlight the corrections introduced.

**Note 2:** The line numbers provided here correspond to the revised version of the manuscript.

GENERAL COMMENT:
The preprint "Importance of ice elasticity in simulating tide-induced grounding line variations along prograde bed slopes" by Maslennikova et al. investigates the impact of viscoelastic processes to establish relations between grounding zone width and ice speed, ice thickness, and bed slope. The authors use a combination of SAR satellite data and a numerical model in their work.
This work presents significant and novel knowledge, building on recent efforts to better understand the impact of elasticity on processes occurring at the grounding zone. I have very few comments regarding the content and methodology of this paper, which I think is of high quality and surely required a lot of work. However, I have several concerns about how methods, findings are presented and discussed, which, in my opinion, need further work before this preprint can be published. Overall, my biggest concern is that this paper lacks a robust discussion regarding the physics of what is modeled, which is briefly mentioned in the conclusion section. There is also some discrepancy in the various sections, where methodology is provided in the results section and a proper discussion is only included in the conclusion section. I think a rearrangement of these sections would really be beneficial to this preprint.
I've made generic and specific comments below to this point, which, in my opinion, would further strengthen the manuscript.

**We thank the Reviewer for the valuable feedback. We believe the implementation of the Reviewer's comments has significantly improved the quality of our manuscript. Below, we provide a line-by-line breakdown of the Reviewer's comments, and a description of the corrections made in response to them.**

**INTRODUCTION**
I find that the introduction contains a lot of information, but at times, it is unclear how this benefits the paper's goals.

**We have rewritten the introduction focusing on how the provided information benefits the paper's goal.**

The authors include many references after each statement, which makes the reading experience slow and confusing. I recommend citing only papers that are directly related to the context. Furthermore, a researcher outside of the peer review team even commented that a couple of citations (Dempsey et al. and Beldon & Mitchell) are completely off-topic. I looked at these papers and agree that they are not relevant to this work, but please correct us if we are wrong. Lines 37-38 cite 16 references (!), and I am not sure how some of these are relevant to the sentence they are attached to.
**Agreed, we acknowledge the Reviewer's concern that some references may appear excessive. As a result, we have revised the reference list:**
- **Line 31-31: (Dempsey et al. and Beldon & Mitchell) citations were removed**
- **Line 34: 16 references were revised and three of the most relevant citations were left to give the reader an idea about the most recent advances in the field:**
  (Cornford et al., 2020;  Gagliardini et al., 2016;  Seroussi et al., 2014; )
- **As the Reviewer suggested, now we cite only the references directly related to the context and ensure every statement has no more than 3 references (except lines 37-38 where all 5 references are necessary to provide a comprehensive literature review):**
  - **Line 25: (Friedl et al., 2020; Haseloff and Sergienko, 2018; )**

- o **Line 27:** ( Davison et al., 2023; Holland, 2008; )
- o **Line 29:** (Goldstein et al., 1993; Schoof, 2007; )
- o **Lines 31-32:** (Albrecht et al., 2006; Coleman et al., 2002; )
- o **Line 34:** (Cornford et al., 2020;  Gagliardini et al., 2016;  Seroussi et al., 2014; )

**DATA and METHODS**

I am not an expert in double differences to examine grounding zone width, but this section leaves me wondering how you can estimate the minimum and maximum grounding zone extension within a tidal cycle if you have only one day of repetitive acquisitions. If you have 24hours of difference between acquisition, wouldn't the tide level being approximately the same. I may have completely misunderstood this section, and if so, I sincerely apologize. However, I still think that even someone without a background in differential interferometry (like me) should be able to quickly understand the physical processes by reading the methodology of this paper. I am aware that substantial past work has used 1-day repetitive acquisitions to study grounding zone migration, and I know that access to sub-daily SAR images is challenging. I am not doubting the quality of this approach; I would simply appreciate a bit more background on this.

**We agree with the reviewer. By using different orbits (ascending and descending) we can acquire data at different times of the day and increase the sensitivity to the tidal spectrum, particularly to the 24-hour tidal cycles. Calculating interferograms from orbits acquired at different times allows us to capture a broader range of tidal frequencies. This statement has been added to the updated version of the manuscript (lines 149-150).**

**VISCOUS AND VISCOELASTIC MODELS**

I appreciate the background information in the modeling perspective, but I am unsure what Equations 1-5 and 9 really contribute to this paper. These are mostly large simplifications of any dynamic system subjected to boundary conditions, and Equation 9 represents a simple tolerance criterion. While I understand that this may be a matter of personal preference, I would consider removing them or perhaps substituting them with the actual PDEs that are resolved (viscous vs. viscoelastic, i.e., Appendix A, eq A1-A3 and A6), which would enrich the reading experience with theoretical background. Equations 1-5 and 9 could go in Appendix A.2.3, where the authors describe boundary conditions. On the other hand, Equations 6-8 are important and useful to the reader. Again, I want to stress that this may be a personal preference, but I do think that Equations 1-5 and 9 can be easily summarized in the text.

**We greatly appreciate this comment, which closely aligns with the opinion of Reviewer 2. To address the concerns of both reviewers, we have revised the 'Viscous and Viscoelastic Models' section (lines 193-365) and moved the model formulation from the Appendix to the main text.**

**We have not changed the contents of Table 1 from the original manuscript, as we believe it provides a concise summary of the model formulation. Additionally, we find it useful to retain this table because it informs the reader about the computational time required for both models. However, after relocating the model formulation to the main text, we felt it was unnecessary to keep the table in the main text. Therefore, Table 1 has been moved to the Supplementary Materials (now Table S4 in the revised manuscript).**

**RESULTS**

This section is really hard to read. There are a lot of numbers, which makes it confusing. I also find hard to distinguish whether model results, model set-up and observational data are discussed. Would dividing this section (data, model) into two subsections help? Finally, authors discuss the simulation set-up in this section, although it should rather go in the methodology section. Results section should only present the outcome of the simulations and not details about simulations themselves.

**We have significantly rewritten the content of this section. We wrote new subsections including measured glacier's parameters, modelled glacier's parameters (and separate subsections addressing the role of thickness, velocity, grounding zone widths and bed slope) and the evaluation of model performance.**

**DISCUSSION**

This section is also quite challenging to follow. In my opinion, this looks more of a results section rather than discussion.

**We have significantly revised the contents of this section.**

I am completely lost in paragraph 272-286, and it took a while to actually understand the series of inequalities that are reported here. These are much more easily visualized in figure 3, and I am not sure if verbatim reporting them is any beneficial.

**Agreed, we removed the set of inequalities in the updated version of the manuscript.**

I think that figure 3 is perhaps the most important, but it is also hard to read, since the y-axis changes for each sub-plot. The reader cannot compare results from viscous and visco-elastic model if the y axis is different. Would a normalized GZ width (0-1) improve things? After doing so, it should be easier to merge panels a-l with panels m-x and improve the figure.

**We tried to normalize the results in Figure 5 (formerly Figure 3) as the reviewer suggested, but we realized the figure did not properly convey the information (see normalized figure below).**

[Figure]

**Normalized Figure 5 (former Figure 3).** The grounding zone values generated by the viscous model (subplots (a) – (l)) were normalized using the viscous model's outputs for a 0.05% bed slope, while the grounding zone values generated by the viscoelastic model (subplots (m) – (x)) were normalized using the viscoelastic model's outputs for a 0.05% bed slope.

**Both models show that for bed slopes between 1% and 5%, glacier thickening affects the grounding zone by less than 1 km, while significant impacts are observed at 0.05% and 0.1% slopes (see Table S3). Based on this, we decided to exclude most of the slopes and retain only 0.05%, 0.1%, 0.5%, and 5% slopes as the most impactful (see Figure 5, line 430). This way, Figure 5 now shows that: 1) the viscous model predicts grounding**

**zones twice as large as those of the viscoelastic model; 2) the evolution of grounding zone width versus glacier thickness is dependent on bed slope; 3) the grounding zone widens with decreasing bed slope. Additionally, the marker shapes were changed to make the figure colorblind-friendly.**
**We also provide the results for all the slopes in the supplementary material (Figure S5).**

**CONCLUSIONS**

There is a lot of information presented here for the first time rather than in the previous sections. The conclusion section should only wrap up the work and draw final arguments.
**We have revised the Conclusions section to only include the summary of the work done.**

As far as I can see, paragraphs 414-421 and 451-460 provide the only physical explanation of what is presented in the paper. I think this work needs a bit more discussion about the physics behind the modeled processes. Why is elasticity so important? It improves model-data agreement, but why? What is the physical reason? I may have missed something, and I apologize if I did, but the only explanation I could find in the text is: "Therefore, an element responsible for rapid deformations, or an elastic component, becomes necessary." To strengthen this paper, I would recommend a thorough discussion regarding the physics of elasticity applied to grounding zone migration. It looks like the model used is based on a previous publication (Stubblefield et al., 2021), so technically, this is not a presentation of a new model. If this is the case, I would appreciate more background on the physical explanation of why this modeling effort is conducted. Furthermore, these considerations should go into a discussion section rather than the conclusion itself.
**Agreed, in the results section we have now included a section titled 'Role of Elasticity'. While viscous models may be adequate for modeling long-term ice sheet evolution, they fail to represent important short-term phenomena such as tidal motion, seasonal cycles, and calving processes, which require the consideration of elastic responses. We also support our argument with existing cryosphere literature discussing this aspect.**

**SPECIFIC COMMENTS**

Line 26: Davis et al 2023, how is this recent paper related to the sentence? Davis et al 2023 does not investigate glacier stability. Also, what does 'salient' mean here? This sentence and pretty much all of the following cite a lot of papers (line 31, 35, 38), which makes the reading experience very slow and confusing at times,
**We removed this reference and revised the reference list.**

Line 48: Gadi et al 2023. This paper investigates ice shelf melting using a numerical model. How is this paper related to "quantification of grounding zone width"?
**We removed this reference and revised the reference list.**

Line 52, Chen 2023, Chen 2023a and Chen 2023b are the same paper. Please consolidate.
**Corrected, thank you.**

Line 63: Please remove parenthesis when you are using a reference as a noun.
**Corrected, thank you.**

Line 162: Really nice figure.
**Thank you. In response to Reviewer 2's suggestions, we revised the original figure and split it into Figures 1, 2, and 3.**

Line 228, 236: This information is not a result but rather an explanation of the simulation setup.
**This information was moved to the newly created 'Model setup' section (lines 296-365).**

Line 260: These are results. I think the logical structure of this paper needs to be revised.
**The logic of the paper was significantly revised and the information the Reviewer is referring to was mode to the 'Results' section.**

Line 275: This and the following lists of inequalities are hard to follow, but easily visualized in figure 3. Is it really necessary to write them down here?
**As mentioned above, the inequalities were removed**

Line 288: Figure 3. I think this is an important plot, but the different limits on the y-axis make it hard to compare between simulations. Would using a normalized grounding zone scale help? Additionally, consider adding this plot in the supplement. Also, please use a color scheme that is colorblind-friendly or, alternatively, different marker shapes.
**Considering the importance of the figure the Reviewer is mentioning here, we did not move the figure to the Supplementary. However, following the Reviewer's suggestions, we significantly revised the figure.**

Line 313: I have a philosophical issue with the term 'validation.' I do not think that you can validate a numerical model; you can at best evaluate how well it agrees with observations. If this model works well in the area of interest, how can you be sure that it is 'valid' for other regions as well?
**Agreed, Section name 'Model validation with DInSAR grounding zone measurements' was changed to 'Evaluation of model performance using DInSAR grounding zone measurements'.**

Line 397: this reads more as a discussion rather than a conclusion.
**This section was revised and a significant part of it was moved to the 'Discussion' section.**

Line 451: This paragraph is the only part of the paper that is an actual physical discussion on the importance of elasticity. I assume that the model used here was already presented in another paper. This work is therefore an extension and an important application of an existing model. In the discussion section, I was expecting a thorough discussion on the theoretical meaning and implications of including elasticity in modeling of grounding zone dynamics, which, alas, is missing. I do not think that the discussion needs to be completely re-written, but some further physical explanation of what is novel here and the overall importance of these findings would really strengthen this manuscript.
**As noted above, we expanded our discussion on the elasticity importance (lines 513-546) and mentioned the physical explanation of elasticity importance in the Conclusions (lines 548-566).**

**We thank the Reviewer for the valuable feedback and look forward for receiving further comments.**

---

## Author Comment (AC3)

**Response to Reviewer 2**

**Blue font:** comments from Reviewer 2
**Black bold font:** Authors' responses to the reviewer's comments

**Note 1:** The Reviewers recommended major revisions, including restructuring the text and redoing certain measurements. The paper is almost entirely rewritten and highlighting these changes would significantly hinder the manuscript's readability. Therefore, the revised version does not highlight the corrections introduced.

**Note 2:** The line numbers provided here correspond to the revised version of the manuscript.

Review of "Importance of ice elasticity in simulating tide-induced grounding line variations along prograde bed slopes" by Maslennikova, et al.
This manuscript compares tidally induced grounding zone width observations to viscous and viscoelastic models to show the viscoelastic model better aligns with observed grounding zone widths. The authors calculated grounding zone widths at Totten, Moscow University, and Rennick glaciers using differential interferometric synthetic aperture radar (DInSAR) by finding the along-flow difference in grounding line location between image pairs taken at two different tide heights and implemented a 2D model of ice response to tides for parameters representative of these three glaciers. The model showed wider grounding zones for shallower bed slopes and wider grounding zones for thicker glaciers, consistent with observations. Furthermore, the model predicted that slow glaciers on steep slopes and fast glaciers on shallow slopes cause the grounding zone to respond most to ice thickness changes.
This paper represents a significant contribution toward modeling tidally induced grounding line variation by providing key comparisons to observed grounding zone widths and recommending a viscoelastic rheology be used in future modeling efforts. The formation of wide grounding zones from tides is an important process impacting basal conditions and ice dynamics, but is often neglected in larger-scale ice flow models. A better understanding of this process, such as from modeling-observation studies like this paper, is essential for more realistic modeling at this critical transition zone. However, the current structure of and analysis in the manuscript obscures this important conclusion, resulting in a difficult to read and difficult to digest manuscript. Below, we highlight four main issues, including that: the main results are hidden across multiple sections (including the appendix), making it harder for readers to understand the key messages; the revised model is not fully described; the InSAR processing does not provide enough details to reproduce or interpret the mapped grounding lines; and the presented figures and tables are often hard to interpret, decreasing their effectiveness. Because addressing these issues will require substantial reworking of the manuscript, we have opted to not list line-by-line minor comments.

**We appreciate the Reviewer's feedback and have made our best efforts to address the provided concerns. Below, we provide a detailed explanation of the corrections made in response to each comment.**

**Issue 1: The structure of the paper hides the key contributions of this manuscript**
For example, methods used in the manuscript appear in Section 2 ("Data and Methods"), Section 3 ("Viscous and viscoelastic models"), Section 4 ("Results"; e.g., lines 211-227 cover how model parameter space and mesh size were determined; lines 238-247 contains more methodological description of the modeling effort), Section 5 ("Discussion; e.g., lines 328-336 describe filtering methods), and Appendix A ("Glacier modeling").
**We have significantly revised the manuscript structure. The model utilizes glacier thickness, ice flow velocity, and bed slope as input parameters. Then, we use DInSAR-derived grounding zones to compare the input parameters with model outputs. For this reason, the 'Data and Methods' section is now structured as follows:**
1. **Brief description of the studied glaciers;**
2. **Explanation of the profiles selection process based on flow velocities and calculation of the ice velocities corresponding to each profile.**
3. **Calculation of ice thicknesses and bed slopes along the profiles;**
4. **Grounding zone width measurements along the selected profiles.**

**We have also rewritten or reallocated entire sections. The statements mentioned by the Reviewer describing the modelling efforts were moved into the 'Viscous and Viscoelastic models' section (lines 193-365):**
- **Lines 211-227 of the original manuscript, which described how model parameter space and mesh size were determined (previously in the Results section), have been moved to the 'Viscous and Viscoelastic Models' section.**

- **Lines 238-247 of the original manuscript, containing a methodological description of the modeling effort (previously in the Results section), have been relocated to the 'Viscous and Viscoelastic Models' section.**
- **Lines 238-247 of the original manuscript describing filtering methods have also been moved to the 'Viscous and Viscoelastic Models' section.**
- **The Appendix of the original manuscript has been revised and integrated into the 'Viscous and Viscoelastic Models' section.**

In fact, the substantial contribution of the manuscript—a new viscoelastic model that can capture a tidally variable grounding zone—is largely relegated to the appendix rather than the methods section of the main text.
**We thank the reviewer for the valuable feedback and agree that the formulation of the viscoelastic model is a significant contribution of our research. We have revised the Appendix section and incorporated it into the 'Viscous and Viscoelastic Models' section (lines 193-365).**

Results from the methods described in the manuscript appear throughout sections 4 ("Results"), 5 ("Discussion"; e.g., lines 261-271), and 6 ("Conclusion"; e.g., lines 408-413).
**The paper structure was revised and most of the text rewritten. The modeling-related information was moved to the 'Viscous and viscoelastic models' section (lines 193-365).**

In addition, the conclusions (Section 6) take the manuscript in a largely unrelated direction, ending with grounding line retreat estimates over multiple decades—a topic that is explicitly not the focus of the manuscript.
**We believe that the section the reviewer is referring to (the last paragraph of the Conclusions section in the original manuscript) offers valuable insights for the cryosphere community, which is why we consider it important to retain. However, we agree with the Reviewer that it is not directly related to the conclusions of the research. Therefore, we have moved the section to the Results section (lines 366-488).**

With information scattered throughout the manuscript, it takes multiple reads to piece together the full scope of the work.
**We apologize for this inconvenience. We have significantly revised the manuscript structure.**

Perhaps more importantly, with so much jumping between topics and concepts, there is no deeper discussion of the physical implications of the data/model comparison. For example, if the viscoelastic model is better than the viscous model because of the short time-periods involved, how does a fully elastic model compare (i.e., why not save the computation time and just implement an elastic model for realistic grounding zone width estimates as suggested by Warburton et al., 2020)?
**Agreed, we have now included a section titled 'Role of Elasticity,' dedicated to discussing the significance of elasticity and explaining why we believe a viscoelastic model better captures glacier dynamics compared to a purely elastic model (lines 513-546). Additionally, we have addressed the physical importance of elasticity in the Conclusions (lines 548-566).**

Does cross-flow heterogeneity matter for the data/model comparison (in other words, does Rennick match the best because it is the simplest geometry or another physical reason)?
**Crossflow heterogeneity can play a role, as the grounding lines may not be perpendicular to glacier flow. Here we have tried to include only grounding lines perpendicular to the glacier flow. We mentioned this aspect in the text (lines 103-104).**

What are the physical reasons for the trends in Figure 5c? (note: Figure 5 is also only first introduced in the Conclusion section, which highlights this structural issue since this section should not contain new analysis). What is the physical reasoning (or even a hypothesis) for why slow glaciers on steep slopes and fast glaciers on shallow slopes cause the grounding zone to respond most to ice thickness changes (lines 284-286)?
**Models and data indicate that wider grounding zones are found where glaciers are thicker. This result can be associated with the increase of the flexural wavelength of ice when its thickness increases (Freer et al., 2023). For thicker ice, the same tidal amplitude affects a larger horizontal distance, leading to a broader grounding zone. This effect is more pronounced on shallow slopes, where the tidal amplitude influences a larger area.**

**Glacier velocity significantly impacts grounding zone width for bed slopes below 0.1% due to the increase in elastic stresses with faster glacier flow (Christmann et al., 2021). As a result, the elastic stress of fast-flowing glaciers on shallow slopes is higher than that of slower-moving glaciers, making the former more sensitive to thickness changes.**

**This figure is now moved to the 'Discussion' section and the 'Conclusions' section was revised. We have included the physical explanation of the observed dependence (fast glaciers on shallow slopes are more sensitive to thickness change than slow glaciers) – lines 491-501**

How do DInSAR measurements at these glaciers validate the model for other glaciers (suggested in the Abstract in line 20), especially with the high variability observed at just these three glaciers?

**The statement was removed from the Abstract.**

We also note that the lack of "flow" in the writing of the manuscript sometimes inhibited our ability to make connections throughout the paper. For example, the model run-time discussion (lines 253-258) interrupted a clean transition from results to discussion (and likely could be included in supplementary material rather than the main text) and the "discussion" of slope coefficients (line 275-285) was quite difficult to comprehend even after multiple tries.

**The paper has been significantly revised. The model run-time discussion is now in a newly created 'Model setup' section (lines 296-365), which summarizes all the modeling efforts.**

Overall, the entire manuscript should be revised with a close eye to structure at all scales—from making sure individual sentences are structured appropriately (e.g., lines 34-35 after the semicolon are a sentence fragment) to ensuring that information in an individual paragraph flows from the topic sentence to making sure all of the introduction is in the introduction, methods are in the methods, and so on.

**Agreed, we have significantly rewritten the manuscript.**

**Issue 2: There are multiple conceptual problems with the model that are un-/under-described**

As suggested above, the main message of the manuscript is the appropriateness and usefulness of the viscoelastic model (compared to a viscous model), but the changes from Stubblefield et al., (2021) to make the model viscoelastic are relegated to Appendix A. Any changes from the Stubblefield et al., (2021) model are key contributions and therefore should be placed in the main text (e.g., Equations A6-A10), which can be guided by Table 1 (a table that likely can be removed if the modeling methods are appropriately described in the main text).

**The information from the Appendix was revised and placed to the main text to the 'Viscous and viscoelastic model' section.**

**We have not changed the contents of Table 1 from the original manuscript, as we believe it provides a concise summary of the model formulation. Additionally, we find it useful to retain this table because it informs the reader about the computational time required for both models. However, after relocating the model formulation to the main text, we felt it was unnecessary to keep the table in the main text. Therefore, Table 1 has been moved to the Supplementary Materials (now Table S4 in the revised manuscript).**

Beyond the lack of description of key modeling efforts in the main text, there are additional conceptual concerns with the model that are not resolved. Below is a subset of the details missing from the model description that fundamentally impact the fidelity and impact of the results. The modeling efforts undertaken in the manuscript need to both appear in the main text and be more fully described before the full impact of the manuscript can be assessed.

Model spin up occurs over a two-month period with water-level set to low tide (lines 243-246, which we note appears in the Results section): why did you choose low tide? Is that the "neutral" stress state? Does choosing a zero-tide or high-tide condition for spin up change the results of the modeled grounding zone width?

**We apologize for the confusion caused by the sentence wording. In the context of the Reviewer's question, a zero-tide position was used to run the model over a two-month period (not a low-tide position). This zero-tide position was set by Stubblefield et al. (2021) for the originally developed viscous model, and we did not change it for the viscoelastic model.**

**The Reviewer raised an interesting question: would the grounding zone width change if a different starting tidal level were used? To address this, we conducted additional model runs for different starting tidal levels: at 0 tide, high tide, and low tide (while keeping the same glacier parameters). We found that the grounding zone width did not change with these initial tidal level variations. We included this finding into the model setup description (lines 346-349).**

This section describing model spin up and experiment duration uses phrases like "enhances results accuracy" and that "models adapt and stabilizes [sic]" after 3-5 days of variable tides, so that the grounding zone width during days 5–7 can be used as the modeling deliverable—how is accuracy and model stabilization defined here since these form the key deliverable of the manuscript?

**The choice of a one-week time limit for the tidal problem allows the model to adapt to tidal impacts. In most tidal model simulations, the grounding zone width slightly increases within the first 3 to 5 days with each tide while the models adapt and stabilize afterward. Several test runs, lasting up to 14 days within the modeling framework, were conducted to estimate the impact of the grounding zone width increase during the initial days. These test runs show that the grounding zone width stops changing after the first five days and remains stable, showing no significant variations afterwards. The initial increase occurs gradually, with the initial grounding zone width being, on average, 80% of the final stabilized width, which is reached after 5 days. Therefore, the resulting grounding zone width value for each model run is determined as the average of the grounding zone width values simulated for days six and seven. This aspect was clarified in the revised manuscript (lines 349-357).**

The inflow velocity of the model is prescribed as a constant, but results from across Antarctica show tidally variable ice velocity on time periods from diurnal to fortnightly and semiannual. Is this inflow boundary condition realistic and/or does variable inflow velocity matter to grounding zone width?

**The current model does not account for variable ice velocities at the inflow boundary, which is a significant simplification. However, our parametric analysis reveals that grounding zone width is not particularly sensitive to changes in glacier velocities (Figure 5).**

How was data from BedMachine integrated into this modeling framework? For example, how was bedrock slope calculated? Figure 1 makes it clear that there is not a "single" slope value, but rather it is highly variable. Is the reported value per flowline just the slope between end points? A linear fit to all the topographic data on a flow line? A mean slope after differentiating bed topography along a flowline? Is it a slope over a certain characteristic length scale that impacts ice flow? Why even reduce the bedrock slope to a single number when the true bedrock topography along each flowline could be incorporated into the finite-element model? Does the true topography vs. a simplified version impact the results?

**The process for calculating velocity, slope, and thickness values along each profile has been clarified in the revised manuscript (lines 98-107, and 119-124). Thickness values were extracted along the profiles and the average thickness was calculated for each profile. Bed slopes were determined for each profile using the 500 m-resolution BedMachine Antarctica topographic map by linearly approximating extracted bed elevation values and calculating the slope of the fitted line. Several test runs using true topography demonstrated that it does not significantly impact the results compared to the simplified topography. Therefore, we opted to use simplified bedrock geometry instead of true topography to better represent the range of input parameters rather than focusing on individual profiles.**

The definition of H from Equation 8 is unclear. Is it only the starting glacier thickness at the position of the starting grounding line? Would you want to compare ending thicknesses at the position of the ending grounding line?

**H is a function of time. However, the code does not directly calculate the evolution of H. Instead, it calculates the evolution of the upper and lower glacier surfaces within a tidal cycle. The evolution of H can be estimated as the difference between the positions of the upper and lower surfaces. Upon analyzing the model outputs, we do not observe any significant changes in glacier thickness during a tidal cycle. Therefore, we feel that including these calculations of changes in H as a function of the tidal cycle is beyond the scope of this study.**

**Issue 3: The DInSAR processing and analysis is not fully described, which impacts the quality of the model/data comparison**

Perhaps most importantly, the differential tide-heights of the interferograms are never mentioned in the manuscript, yet the results from these observations are compared to "tides with a 1 m amplitude" in the model (line 177). Since the grounding zone width depends on the tidal amplitude, are the DInSAR results showing a 1 m tidal amplitude (note: it is not clear whether this means ±0.5 m tides or ±1 m tides for the modeling effort)? If they are not, this is not an apples-to-apples comparison between the observed widths and modeled widths (i.e., the overlap between modeling results and DInSAR analysis in Figure 4c may in fact indicate that the model has substantial bias). There are some

good examples showing clear DInSAR differential tide estimates in the literature (e.g., Table 1 in Milillo et al., 2017 or Table S1 in Milillo et al., 2022).
**Agreed. We have added Table S6, which summarizes the Tidal and Inverse Barometer Effect (IBE) components. The glaciers experience 1-meter amplitude tides (2 meters peak-to-peak). Therefore, the model setup utilizes a sinusoidal wave with a 1-meter amplitude. However, after applying the IBE correction, differences in tidal heights between the corresponding pairs of DInSAR interferograms are 0.95 m for MU, 1.03 m for TOT, and 1.08 m for REN (Table S6). To ensure a valid comparison between the DInSAR measurements and the model outputs, the modeled grounding zone measurements are referenced to the corresponding IBE-corrected maximum tide (column H in Table S6) for the DInSAR interferograms. This clarification has been added to the text (lines 156-178).**

The InSAR processor used is not mentioned in the manuscript (and thus the results are not reproducible).
**GAMMA software was used for data generation, which we now mention in the text (lines 139-140).**

The text says uncertainty of manual grounding-line mapping was "empirically determined" to be 200 m (line 228) with no additional explanation. What were the methods for assessing the uncertainty in grounding line delineation?
**We refer to the average accuracy of manual grounding line mapping from a DInSAR interferogram. Performing a grounding line mapping several times over the same interferogram, on average, a human expert may place a grounding line with a 200 m deviation (Rignot et al., 2014; Ross et al., 2024). We have clarified this aspect in the manuscript (lines 154-156, 182-183, and 332-333).**

The DInSAR grounding lines at high and low tide are (unsurprisingly) wiggly in Figure 1 and appear to intersect for Moscow University and Totten—how do you proceed when the low-tide grounding line is upstream of the high-tide grounding line (negative grounding zone width)?
**In these areas with "negative" grounding lines (GL), we did not analyze grounding zones for two reasons: 1) GL are too close to each other (within the manual mapping error) and therefore grounding zone measurements here are insignificant, 2) cross-flow heterogeneity can play a role as the grounding line is not a straight line perpendicular to flow.**

Do the tidal flexure patterns deviate from those observed on the glaciers studied in Rignot et al., (2014) and would that impact your results?
**We did not analyze the tidal flexure patterns in Rignot et al. (2014) as the authors did not provide unwrapped interferograms in either the main text or supplementary materials. Additionally, there is limited availability of grounding line data acquired at different tidal levels within timespans where horizontal flow has changed significantly, making it difficult to incorporate these data into our models. However, while we do have more recent grounding zone data for the Amundsen Sea Embayment (Milillo et al. 2019, Milillo et al. 2022), these areas are primarily characterized by negative bedrock slopes, which we plan to address in a follow-up study.**

Finally, how were the flow lines for determining grounding zone width chosen? They look quite straight; are they the true path of an ice parcel or an estimate? Would choosing different flowlines substantially impact your results? We quickly show that estimated flowlines on Totten Glacier (below, left panel, black lines) do not match the lines in Figure 1 (below, right panel) with substantial differences in flow direction, suggesting grounding zone widths reported in the manuscript may contain substantial unaccounted for error:

[Figure]

**We greatly appreciate the comment and have re-analyzed the data accordingly. We adjusted the direction of the profile lines to align perfectly with the ice flow direction (see Figure 1 and lines 98-107). We recalculated**

**the parameters for the glaciers and the grounding zone widths. The updated results are presented in Figure 6, Figure 7, Figure S1, Figure S4, Table S1, and Table S5. These new results show improved correspondence between the DInSAR-based measurements and the data compared to the original analysis. Therefore, in response to the reviewer's question, 'Would choosing different flowlines substantially impact your results?', the answer is no. While the individual results did change, the average values of all the parameters remained nearly unchanged (see Table S1). As a result, the overall assessment of the models' performance was not significantly impacted.**

**Issue 4:Figures and tables are not of publishable quality.**

For example, Figure 1 does an excellent job at conveying a lot of information, but there are several aspects that make it hard to interpret. There are no figure limits for this journal, so it is not clear why so much information is compressed into one figure. Unpacking the panels into multiple figures will help strengthen your argument as will considering the following issues:

**We have divided Figure 1 into three separate figures: the location of the glaciers of interest in Antarctica and the ice velocity map (Figure 1), the bed topography and the ice thickness map (Figure 2), and the DInSAR interferograms with the grounding zones (Figure 3). We have also addressed the rest of the Reviewer's concerns (see below).**

- Having the same color scale with different ranges makes comparison between glaciers unintuitive. Color scales should be sequential and perceptually uniform covering the same range for all three glaciers where reasonable and explicitly stating if you diverge from this standard to make sure the reader does not misinterpret a panel (as we did initially).
  **In the revised version of the manuscript, the colorbars were made uniform across the glaciers. We made the colorbars by following the recommendations from** *Crameri, F., Shephard, G. E., & Heron, P. J. (2020). The misuse of colour in science communication, Nat. Commun., 11, 5444.*

- Several labels are far too small and hard to read. For example, the numbering of the flowlines is near impossible at 100%.
  **The labels were enlarged in the revised version of the manuscript.**

- Several color choices are difficult to distinguish. For example, the blue dashed line indicating the high (low) tide grounding line is nearly invisible in panel l (o).
  **In the revised version of the manuscript, the colors were modified to improve the clarity.**

- Please label which glacier corresponds to which column.
  **The glaciers were labeled in the revised version of the manuscript.**

- Colorbars in the bottom two rows should not be labeled dz as the colors show a wrapped interferogram. It is not even clear that this is phase projected into the vertical z component, so clarity on this label is important.
  **'dz' was replaced with 'DInSAR phase' in the revised version of the manuscript.**

Figure 3 does not provide a clear message and left us confused. Is there a specific trend that we are supposed to observe? If so, it might be beneficial to extract the panels that show this trend and move the rest to the supplement. If all the panels are important, we suggest adding additional labels showcasing what comparisons are relevant. Additionally, please make the axes limits consistent between plots so the reader can effectively compare between panels.

**We have modified the figure. Both models show that for bed slopes between 1% and 5%, glacier thickening affects the grounding zone by less than 1 km, while significant impacts are observed at 0.05% and 0.1% slopes (see Table S3). Based on this, we decided to exclude most of the slopes and retain only 0.05%, 0.1%, 0.5%, and 5% slopes as the most impactful (see Figure 5). The goal of this figure is to show that: 1) the viscous model predicts grounding zones twice as large as those of the viscoelastic model; 2) the evolution of grounding zone width versus glacier thickness is dependent on bed slope; 3) the grounding zone expands with decreasing bed slope. Additionally, the marker shapes were changed to make the figure colorblind-friendly.**

The horizontal bars in Figure 4 are not "error bars" as described, but rather are a range of modeling results (at least in our interpretation; the description of these bars in the text and caption was confusing).

**Corrected (lines 442-444).**

How are the pink and green outlines calculated in Figure 4 and 5?
**We have modified these figures (now figures 6 and 7) to make this aspect more evident and explained this in the text as well (lines 458-467 and 508-511).**

Why do different glaciers have different bin sizes in the histograms for Figure S3?
**We set the uniform bin width (now Figure S4).**

Table 2 does not explain what these numbers represent (e.g., are these means of the minimum, mean, and maximum of each flowline? What is represented by the ± value?).
**The table has been moved to the Supplementary Materials as Table S1. It presents the minimum, maximum, and mean values for the glaciers, derived from all the flowlines associated with the glacier of interest. We mention this in the table description.**

Also "MeASURESs2" is not a data product— MEaSUREs is an acronym for a NASA funding program (Making Earth System Data Records for Use in Research Environments).
**In the revised manuscript, we specified that every time we mention the MEaSUREs InSAR-based ice velocity map of Antarctica (lines 65, 99, 111, 116, and 130).**

The organization of Table A1 is confusing—some of the geometric quantities are scalars, but there is a separate table heading for scalar quantities? H likely should be H(t), l should be l(t), there are no units provided (which would be quite helpful for the reader), and the scalars are bolded, which may be interpreted as a vector quantity.
**Table A1 has been revised as follows:**
- **The fields 'vector quantities', 'scalar quantities', and 'tensor quantities' have been combined under the heading 'physical quantities'.**
- **We have specified the type of each field (scalar, vector, or tensor) in a separate column, and only vector and tensor quantities have been bolded.**
- **We indicated that H and l are time-dependent by denoting them as H(t) and l(t).**
- **Units have been provided in a separate column.**

The figures and tables all require dedicated consideration to make best use of journal space to support the text.
**The figures and tables have been reviewed and modified where necessary (see above).**

**We hope the revisions to the manuscript address the Reviewer's concerns and look forward to any further feedback.**

---

## Author Response (AR3)

Importance of ice elasticity in simulating tide-induced grounding line variations along prograde bed slopes
*by Natalya Ross, Pietro Milillo, Kalyana Nakshatrala, Roberto Ballarini, Aaron Stubblefield, Luigi Dini*

**Response to Reviewer 1**

**Blue font:** comments from Reviewer 1
**Black bold font:** Authors' responses to the Reviewer's comments

**Note:** The line numbers referenced here correspond to the revised manuscript **with** highlighted corrections.

GENERAL COMMENT:
I would like to begin by thanking the authors for addressing the comments and feedback I provided during the first review cycle. I recognize that this likely required considerable effort, and I appreciate their responsiveness. The manuscript has shown significant improvement since its initial submission. The message is now much clearer, and the reading experience is considerably smoother. The manuscript presents a satellite-derived estimate of grounding zones along the Sabrina Coast in East Antarctica, utilizing DInSAR techniques. The authors also compare their results with a numerical model, examining the elastic and viscoelastic components of ice flow dynamics. I have no major concerns at this stage, but I do have a few minor comments on the figures. Assuming these are addressed, I would recommend the manuscript for publication in The Cryosphere.

**We sincerely appreciate the Reviewer's valuable feedback. We have made every effort to thoroughly address the previous comments and firmly believe that incorporating the current suggestions has further strengthened our manuscript. Below, we provide a detailed response to each comment, along with a description of the corresponding revisions.**

Figure 1: I find the geo-referenced figures a bit confusing. While I understand that the coordinates are expressed in polar stereographic, I wonder if these numbers may be difficult to interpret for readers who are not familiar with this reference system. Would it be possible to present the coordinates in latitude and longitude instead, as this might be a clearer choice? Additionally, I have some concerns about the rectangles in panel (a). I think they should be meant to represent the zoomed-in areas shown in the subsequent panels? It doesn't seem that they do.
Figure 2,3: Although the authors appear to be showing the same zoomed-in area as in Figure 1, the geometry seems distorted due to inconsistencies in the size of the inset boxes. For instance, panel (c) in Figure 1 is noticeably larger than panels (b) and (d), but this size disparity changes again in Figures 2 and 3. In my opinion, maintaining consistent sizing across all figures would help readers more easily recognize that the same area is being depicted throughout. Additionally, the comment I made regarding the coordinate labels in Figure 1 is also relevant to Figures 2 and 3.

**We address the comments regarding the three figures collectively:**

1) **The projection used in all three figures is EPSG:3031, which defaults to meters as the unit. We believe that changing the default units from meters to degrees would be confusing for those familiar with this projection. However, we attempted to convert the coordinates from meters to latitude/longitude and found that it degraded the figure quality (see the modified Figure 1 below). In latitude/longitude coordinates, the grid becomes tilted, making the figure more difficult to interpret.**

Importance of ice elasticity in simulating tide-induced grounding line variations along prograde bed slopes
*by Natalya Ross, Pietro Milillo, Kalyana Nakshatrala, Roberto Ballarini, Aaron Stubblefield, Luigi Dini*

[Figure]

*Modified Figure 2 (formerly Figure 1) with lat/lon coordinates.*

2) **Panel (a) in Figure 2 (formerly Figure 1) displays the locations of the glaciers, as specified in the text (line 80), rather than the contours of the zoomed-in areas.**

3) **Due to the significant differences in glacier sizes, standardizing the figures to the same size and scale reduces their clarity and overall readability (see the modified Figure 1 below). For example, the Rennick Glacier is the smallest of the three, and resizing its subplot would introduce excessive irrelevant surrounding area while shrinking the main trunk. This also makes it significantly harder to label and number the profile lines, as they become too densely packed.**

[Figure]

*Modified Figure 2 (formerly Figure 1) to the same size and scale of the subplots.*

**For the reasons listed above, we have opted not to modify the figures in the revised version of the manuscript.**

Figure 5: I believe this figure is much clearer now. Thank you for your effort in addressing my concerns with its initial version.
**Thank you, we appreciate your comment!**

**We thank the Reviewer for the valuable feedback and look forward for receiving further comments.**

**Response to Reviewer 2**

**Blue font:** comments from Reviewer 2
**Black bold font:** Authors' responses to the reviewer's comments

**Note:** The line numbers referenced here correspond to the revised manuscript **with** highlighted corrections.

Review of "Importance of ice elasticity in simulating tide-induced grounding line variations along prograde bed slopes" by Ross et al.
This revision has greatly improved the manuscript's structure, helping readers understand the importance of your work in the context of tidal grounding line variations. There are still a few major structural points that we think would strengthen the manuscript. Below, we list these points along with minor corrections to consider as you revise.

**We appreciate the Reviewer's thoughtful feedback and have made every effort to address the concerns raised. Below, we present a detailed explanation of the revisions made in response to each comment.**

**Major Comments**

1. The current organization of §2 and §3 makes your workflow opaque. For example, we expect the DInSAR measurements from §2.2.3 to be used as model inputs rather than validation because the section is squeezed between a section describing model inputs and a section describing the model. We think that a flowchart or schematic describing what data is used as input into the model, what outputs the model produces, and how DInSAR measurements are used to validate the model will help readers parse the manuscript.

**Agreed, section 2.2.3 has been renumbered as section 2.3. As requested, we have added schematics of the modeling process as Figure 1.**

2. Your revision to Figure 5 looks great and helps us visualize what is important between the models as grounding zone width changes. You do a great job summarizing these points in your reply to both reviewers, but the description of how glacier thickness, velocity, and bed slope impact the grounding zone in §4.2 is confusing as written. We think it would be beneficial to include the main points in your reply to reviewers in addition to the specific examples given with numbers.

**The main points from section 4.2 are:**

1. *In both the viscous and viscoelastic models, the grounding zone width (GZ) exhibits a linear relationship with glacier thickness (H). The main difference between the two models lies in the magnitude of the modeled grounding zone width as a function of glacier thickness over varying bed slopes.* **(lines 427-428 and 432-433)**
2. *In both models, shallower bed slopes increase the sensitivity of grounding zone width to changes in glacier thickness. The viscous model is more sensitive to ice thickening compared to the viscoelastic mode.* **(lines 436-437)**
3. *The most pronounced effect of velocity changes on grounding zone width occurs at shallower slopes of 0.1% and 0.05%. For these slopes, an increase in ice velocity from 100 m/year to 800 m/year can result in up to a 60% increase in grounding zone width in both the viscous and viscoelastic models.* **(lines 447-450)**
4. *In both models, grounding zone width increases as the bedrock slope decreases, indicating that the relationship between glacier bed slope and grounding zone width follows an inverse power law.* **(lines 466-468)**
5. *For any combination of bedrock slope, glacier thickness, and ice inflow speed, the grounding zone width obtained from the viscoelastic model is nearly half that of the grounding zone width calculated by the viscous model on shorter time scales.* **(lines 473-475)**

**Minor Comments**
*[Line number of manuscript v3 given in brackets if applicable]*

**General comments:**

- Unit abbreviations are inconsistent; please choose m/yr or m/year

**We made the abbreviations consistent: now, we use m/year everywhere**

- Supplement and figures should be cited in the order of first appearance (e.g., Table S6 is first mentioned on line 162, before Tables S2-S5 and Figure 6 is first referenced on line 185, before Figures 3-5)

**We arranged the supplementary tables and figures in the order in which they appear in the manuscript.**

[13] What type of tides does each region have? Does having more time for viscous components of the model to equilibrate matter (e.g., if tides are primarily diurnal versus semidiurnal)?

**In all our tests, we found that a two-month run with a stationary ocean and no tides, followed by a seven-day period with tides incorporated, allows the model to reach stability. Our results align with Stubblefield et al. (2021). We have tested a single semidiurnal sinusoidal tide and acknowledge that results may change if constructive or destructive interference occurs between diurnal and semidiurnal tides. Therefore, we reserve the pleasure of this analyzing different tidal scenarios for a follow up study.**

[19-20] Word choice: what error is considered a match of the model to measurements? ~74% more accurate is not meaningful without context from §4.3.

**Agreed we have modified this sentence with** *'We establish the dependence of the grounding zone width on glacier thickness, bed slope, and glacier flow speed and find that grounding zone predictions using a viscoelastic model significantly outperform those of a purely viscous model.'*

[21] Underscore the critical role played by ice elasticity in continuum mechanics-based glacier models *on daily tidal time scales*.

**Thank you for the advice! We added 'on daily time scales' into the sentence (line 22)**

[37] This sentence does not explain the cool science you have done to make your short-term model include viscous components and better match data! It just seems like you are following the status quo of short-term models.

**Thank you, in the updated manuscript (now lines 37-39), we clarify that our study focuses on short-term models and highlight the 'status quo' – we disregard long-term glacier evolution. The physical aspects of our model are introduced later (lines 59–61) to maintain the logical flow of the text.**

[56] full Navier-Stokes

**We removed the word 'full'**

[60-64] This line seems overly specific for the intro — these are methods and should be moved to the appropriate location.

**Agreed, we removed the sentences the Reviewer is referring to**

[71] "determine" is a strong word here that might imply the development of closed form parameterizations. A word like "assess" or similar is probably better here.

**We replaced 'determine' with 'assess'**

[76] Figure 1 shows locations, not relative locations

**We removed the word 'relative' from the figure description**

[82-84] This context on mass balance cites a paper that is over a decade old. There are much more up to date assessments of mass balance that should be provided.

**We replaced (Rignot et al. 2013) with (Li et al. 2022) – (line 88)**

[84-85] Just being grounded below sea level does *not* make a glacier potentially susceptible to collapse. It requires a reverse slope bed and TOT has ~ 40 km of prograde slope ("This means that the main trunk of TG will not be retreating on a retrograde bed and be prone to a marine instability in the coming years" from Li et al., 2015). Please revise accordingly

**We replaced this sentence with 'Moreover, between 2010 and 2018, both glaciers experienced higher basal melt rates than the neighboring glaciers in East Antarctica due to the intrusion of warm ocean water into their subglacial cavities', which adds two recent publications into the reference list: Adusumilli et al., 2020, and Nitsche et al., 2017 (lines 88-91)**

[89-92] Again, Pritchard et al. (2012) is 13 years old and multiple new, higher fidelity, longer-term estimates of basal melt have been published. It is important to look to the full body of literature when appropriate, but for specific values and assessments of stability, we should be using the most up-to-date perspectives.
**Line 96: we replace Pritchard et al., 2012 with Adusumilli et al., 2020**
**Line 98: we replace Pritchard et al., 2012 with Baumhoeret al., 2021**

[95] How do you choose the 69 profiles? They obviously do not matter much if your results are similar to when you used the nonphysical profiles in the first draft, so maybe mention that exact profile choice only saw ± x% error? You do not choose areas with negative or near-zero grounding zone widths (e.g., the gap between contours 25 and 26 on MU). Why? This looks like an interesting region as no GZ migration is observed. (This issue comes up again on line 102 where it says flow lines "were spaces 500 to 600 m apart" when there is a larger gap on MU)
**Our InSAR-based grounding line has an accuracy of 200 m, resulting in a grounding zone accuracy of 400 m. We did not consider areas where the grounding zone width is less than 400 m, as this would result in an error exceeding the observed value. This explains the gap between profiles 25 and 26. Therefore, the 69 profiles were selected along the main trunks where the grounding zone width exceeds 400 m. We clarified that in the text:**
***'However, due to the 400 m grounding zone mapping error, we did not position the profiles in areas where the grounding zone width is less than 400 m, which resulted in a larger spacing between MU's profiles 25 and 26.'***
**(lines 111-113).**

[98-99] Rignot et al. (2017) has been superceded by the Mouginot et al. (2019) phase-based velocity map, which is considerably higher fidelity. Is there a reason here to use a somewhat out-dated velocity product?
**The reviewer's observation is valid, as the term 'phase-based' map may be misleading. Below, we explain why using Mouginot et al. (2019) would not change our results.**
**The 'phase-based' map from Mouginot et al. (2019) relies solely on InSAR phase data for the slow-moving interior of Antarctica. However, for fast-moving regions, such as the grounding zone we analyzed, it still uses the same pixel-offset-based dataset from Rignot et al. (2017). The Mouginot et al. (2019) map is more accurate only in slow-moving regions. The results remain unchanged in areas where only the pixel-offset technique has been applied in both the Mouginot and Rignot maps. The launch of NISAR or the recently deployed Sentinel-1A/C with a one-day revisit configuration will improve this situation, as it may allow phase-based measurements over grounding line regions of these glaciers.**

[101] The expression "arctan(v_x/v_y)" can only provide direction across 180°, so we hope this was calculated not directly as the quotient, but rather as a two-argument function.
**We used the $\mathrm{np.\,arctan2}(v_y, v_x)$ function for the direction calculation and have clarified the calculation process in the text (lines 108-110).**

[101-102] The phrasing here is a little confusing. "Flow lines were selected along the flow direction" makes it sound like each flow line is selected along flow (which does not make sense since it would just select the same flow line over and over again). Did you mean they were selected across the direction of flow and generated for XX km along flow, centered on the grounding line (or something similar)?
**Corrected, thank you (lines 110-111)**

[104] Parallel --> perpendicular. Also, some of the lines on, e.g., MU are indeed parallel to the grounding line. Crossflow heterogeneity is mentioned as impacting your analysis in lines 103-104, but there is no further mention of what possible errors it may introduce or how you might address those errors in future work.
**Thank you! We replaced 'Parallel' with 'perpendicular'**
**We extended the discussion on crossflow heterogeneity: 'Since the model does not rely on individual measurements but rather on the range of observations to ensure comprehensive coverage of the considered glaciers, crossflow heterogeneity does not impact the results of our analysis' (lines 360-362)**

[109] Figures 1 and 2 still have confusing colorbars. For example, the Figure 1 colorbar has two yellows and two pinks that look nearly the same. We had a hard time determining that Totten is up to 400 m/y faster than Moscow University from a quick look at the figure (until we saw the contour lines).
**We have modified the colorbar in Figure 2 (formerly Figure 1) and removed repeating colors. We did not modify Figure 3 (formerly Figure 2) as it does not have repeating colors. Moreover, we consulted with a**

**colorblind person, who confirmed readability of both figures. We acknowledge that color blindness exists in different forms and have made our best effort to accommodate the reviewer's request.**

[122-123] "linearly approximating extracted bed elevation values": does this mean you fit a linear polynomial to the full profile and positive indicates prograde (i.e., higher upflow)? This is a little counterintuitive as we would reflexively think in an upflow-to-downflow frame, so a prograde slope should be a negative number), which just highlights that the reference frame should be defined and how this bedrock slope estimate is calculated should have a bit more clarity. Moreover, there seems to be a bit of a mismatch between the Figure 2 and Table S1 in that there appear to be some retrograde bed slopes (i.e., negative), particularly along the right side of MU, yet Table S1 indicates there are no negative bed slopes. This also plays back into the comment on lines 84-85, where the majority of these grounding line bed slopes are prograde, not retrograde, indicating there is *not* a substantial concern (at least in the chosen locations) of instability.

**We thoroughly rechecked the slope values for profiles 30–33 over MU (we believe the reviewer is referring to these). Considering the uncertainties associated with BedMachine data and the use of a linear approximation, these profiles exhibit very shallow but positive slopes (please refer to Figure 7, where the reviewer can see four very shallow slopes of 0.02% with grounding zones extending over 5.5 km). Moreover, even if we exclude these four profiles from consideration, it will not change the conclusions we make.**

**We understand that reviewers may find the classification of these positive slopes as prograde counterintuitive. However, assigning positive and negative slope values to prograde and retrograde terrains, respectively (or higher/lower bed elevations upflow), aligns with the Latin origins of the terms 'pro' and 'retro,' meaning 'in front of' and 'behind.' Positive numbers are greater than negative numbers, meaning they come 'in front of,' while negative numbers stay 'behind.' In other words, our convention follows a downflow-to-upflow framework, which is widely adopted in the cryosphere community:**

1. **Rignot, E., 2022. Sea level rise from melting glaciers and ice sheets caused by climate warming above pre-industrial levels. Phys.–Uspekhi, 65(1).**
2. **Rignot, E., Ciracì, E., Scheuchl, B., Tolpekin, V., Wollersheim, M. and Dow, C., 2024. Widespread seawater intrusions beneath the grounded ice of Thwaites Glacier, West Antarctica. *Proceedings of the National Academy of Sciences*, *121*(22), p.e2404766121.**
3. **Li, X., Rignot, E., Morlighem, M., Mouginot, J. and Scheuchl, B., 2015. Grounding line retreat of Totten glacier, East Antarctica, 1996 to 2013. *Geophysical Research Letters*, *42*(19), pp.8049-8056.**
4. **Sergienko, O.V. and Wingham, D.J., 2022. Bed topography and marine ice-sheet stability. *Journal of Glaciology*, *68*(267), pp.124-138.**
5. **Schoof, C., 2012. Marine ice sheet stability. *Journal of Fluid Mechanics*, *698*, pp.62-72.**
6. **Milillo, P., Rignot, E., Rizzoli, P., Scheuchl, B., Mouginot, J., Bueso-Bello, J.L., Prats-Iraola, P. and Dini, L.J.N.G., 2022. Rapid glacier retreat rates observed in West Antarctica. Nature Geoscience, 15(1), pp.48-53.**
7. **Batchelor, C.L., Christie, F.D., Ottesen, D., Montelli, A., Evans, J., Dowdeswell, E.K., Bjarnadóttir, L.R. and Dowdeswell, J.A., 2023. Rapid, buoyancy-driven ice-sheet retreat of hundreds of metres per day. *Nature*, *617*(7959), pp.105-110.**
8. **Chen, H., Rignot, E., Scheuchl, B. and Ehrenfeucht, S., 2023. Grounding zone of Amery ice shelf, Antarctica, from differential synthetic-aperture radar interferometry. *Geophysical Research Letters*, *50*(6), p.e2022GL102430.**

[126] Remove "relief"
**Removed**

[145-146] Horizontal flow velocity is also not uniform even on tidal timescales – what error is introduced by assuming the double difference interferogram cancels out the horizontal deformation (e.g., Rack et al., 2017; Wild et al., 2019)?
**In our double-difference interferograms, if there were uncompensated horizontal deformation, it would appear as a distinct fringe pattern in our data. We do not observe any characteristic pattern associated with residual horizontal deformation, indicating that the potential error introduced is negligible. No changes are needed in response to this comment.**

[158-161] Why is the IBE correction applied here? Doesn't IBE contribute to grounding line motion as well?

**The IBE is used to calculate the effective tidal height in our DInSAR interferograms to ensure that when comparing DInSAR grounding zone measurements with tidal model outputs, we select the corresponding tidal height in the models. This is specified in the manuscript at lines 189-192:** *'To ensure a valid comparison between the DInSAR-derived and modeled grounding zones, the IBE-corrected tidal levels listed in column H of Table S6 are used to evaluate the grounding zone width in the model. The modeled grounding zone widths are measured by calculating the difference between the grounding line position at maximum H and minimum H.'* **The IBE is not included in our model, and we plan to analyze this aspect in future studies.**

[166] Why can you assume that the grounding line position observed in the interferogram corresponds to the largest tidal level among the acquisitions? Does this mean you cannot image low tide very well and are missing important information about GL migration near low tide?

**High tides facilitate water infiltration upstream (Rignot et al., 2024); therefore, the highest tide in our double-difference InSAR acquisitions represents the mapped grounding line. The reviewer is correct that, under this assumption and with this approach, we will never be able to capture the grounding line at the lowest tidal level. We have added this clarification on lines 180-181:** *'Following this assumption our approach will never be able to map the grounding line at its lowest tidal position.'*

**Rignot, E., Ciracì, E., Scheuchl, B., Tolpekin, V., Wollersheim, M., & Dow, C. (2024). Widespread seawater intrusions beneath the grounded ice of Thwaites Glacier, West Antarctica. Proceedings of the National Academy of Sciences, 121(22), e2404766121.**

[170-172] Thank you for including the table of height changes from DInSAR measurements for each glacier (though we note these are results, not methods, so should probably be moved).
We noticed there is only one set of measurements (high and low tide) for each glacier (as a result of tasking constraints we presume), which do not always span a representative range of tides. Because this is tasked and there does not seem to be more available data, some discussion of the limitations of the analysis given the limited dataset is critical later in the manuscript.

**Agreed, we have added the following clarification on lines 523-527: '***As shown in equation (44), a one-meter amplitude semidiurnal tide (12-hour period) was used to simulate the tidal behavior of the studied glaciers. It should be noted that our DInSAR grounding line measurements rely on a single set of tidal observations (high and low tide) for each glacier. While these measurements do not always cover the full tidal range, they consistently capture a differential tidal variation of approximately one meter (Table S2).***'**

[202] You do an excellent job explaining model variables as they appear, so we think having Table 1 in the main text now detracts from the read-through. We would put it in an appendix or the supplement for easy reference.

**We moved Table 1 to the Supplement.**

[205] Do you want to use (x,z) to be consistent with Stubblefield et al., (2021)?

**We find the (x,y) coordinate system more appropriate in this context, as the (x,z) system may give some readers the impression of a three-dimensional representation. Therefore, while we appreciate the Reviewer's suggestion, we have chosen to retain the (x,y) system without modifications.**

[211] Line 211 and equations 3,4,5 should replace "f(x)" with "b(x)" to be consistent with Figure 4 and Equation 41 (or use β(x) as in Stubblefield et al., 2021).

**We replaced "f(x)" with "b(x)"**

[234] We might use η(D) rather than η(v) in equations 8 and 10 (local η should only depend on invariants of D). This is also consistent with using η(τ) in equation 13.

**We replaced η(v) with η(D) in equations 8 and 10.**

[250] Shouldn't this definition come after eq. 12 and not eq. 13?

**We moved the definition to come after eq.12.**

[279.5] Why do you jump around from indicial to tensor notation in equation 30?

**Depending on the mathematical operations performed, indicial notation may be more advantageous than tensor notation, and vice versa. We believe that our derivation approach is both efficient and accessible to readers with the relevant background. Therefore, we respectfully choose not to modify the derivation process.**

[319] Does adding additional tidal constituents change the results?

**We have not analyzed additional tidal constituents or IBE effects as we believe this could be an excellent starting point for future studies.**

[322] Thank you for clarifying that the data-model comparison is not one-to-one. We interpret the comparison as selecting points from the model at the same tidal level as the interferogram, but the wording is confusing. We suggest revising this section with an emphasis on the precise comparison made. Also, if you are modeling 1 m tides, then are subselecting, is this a fair comparison if the tidal range is different from 2 m here? In other words, the model is missing the maximum stresses if the modeled tide range is different from the actual tide range at these glaciers (which is important for the viscous component).

**We understand the reviewer's confusion; however, we are not missing the maximum stresses in the model, as the tidal amplitude remains constant for each run. We simply sample the model grounding line position at the DInSAR-measured tidal heights. We have revised our wording for clarity and now state on lines 339-344: '***To ensure a meaningful comparison between the measurements and the model results, the modeled grounding zones were calculated using a 1-meter tidal amplitude while considering the model grounding line position at the DInSAR tidal heights (Column H in Table S2). In other words, the high- and low-tide sea levels derived from the interferograms were used to extract the modeled grounding line positions and subsequently determine the corresponding grounding zone.***'**

[376] "%" --> "m/yr"

**Replaced '%' with 'm/year'**

[382-383] The statement, "MU introduces some variability, particularly due to its wide grounding zones exceeding 6km", is confusing. On Figure S1, there does not appear to be any MU grounding zones that exceed 6 km in width at all. In fact only Totten has grounding zone widths that exceed 6 km.

**Thank you for the observation! We replaced 'MU introduces some variability, particularly due to its wide grounding zones exceeding 6km' with 'MU introduces some variability, particularly due to presence of narrow grounding zones under 1 km.' (lines 401-402)**

[393-394] Why is this assumption being made? The data collection time from 1996 should be available and therefore the tide stand of that grounding line can be explicitly estimated

**Indeed, the acquisition time of the 1996 data should be available but is not accessible to the authors. Furthermore, the cross-comparison of C-band and X-band grounding lines, along with their respective accuracies, is limited and beyond the scope of this study. Therefore, we do not find it appropriate to extend our analysis to include the 1996 tidal levels. We clarified this aspect in the text (line 412)**

[438] Y-axis does not show evolution, but rather the size of GZ for different chosen ice speeds.

**Changed 'while the y-axis shows the evolution of the grounding zone as the glacier becomes thicker' to 'while the y-axis shows the grounding zone width.' (lines 458-459)**

[442] What does "where the results for different inflow speeds are averaged by glacier thickness" mean?

**We clarified the sentence by replacing 'where the results for different inflow speeds are averaged by glacier thickness' with 'where the grounding zone values, indicated for different glacier thicknesses, are obtained by averaging the grounding zones for the same thickness across varying inflow speeds' (lines 462-463)**

[501] "confirms" is not a great word choice here. It is certainly consistent with your observations, but other processes can account for this apparent sensitivity.

**Replaced 'confirm' with 'support' (line 523)**

[502] Figure 7: What justification do you have for the glacier thickness/flow speed/GZ length dependence? We could draw a line with a positive or negative slope within the error bars presented in Figure 5c.

**We acknowledge that we do not have extensive data to unambiguously fit a linear relationship. However, our justifications are based on literature studies, as discussed in Section 5.1 of our manuscript. Our findings can be summarized by the following key points:**

1. **GZ Wider where glaciers are thicker (lines 517-520):**
   *This observation can be associated with the increase of the flexural wavelength of ice when its thickness increases (Freer et al., 2023). For thicker ice, the same tidal amplitude affects a larger horizontal distance, leading to a broader grounding zone. This effect is more pronounced on shallow slopes, where the tidal amplitude influences a larger area.*
2. **GZ Width for Shallow slopes and glacier velocity (lines 520-523):**
   *Glacier velocity significantly impacts grounding zone width for bed slopes below 0.1% due to the increase in elastic stresses with faster glacier flow (Christmann et al., 2021). As a result, the elastic stress of fast-flowing glaciers on shallow slopes is higher than that of slower-moving glaciers, making the former more sensitive to thickness changes and supporting our observations.*

[530-532] This line is off the mark. Plenty of people have used pure elastic models successfully to model flexure over tidal timescales and compare to observations. Yes, you will not fully capture all glaciological processes, but you can certainly investigate individual processes with elastic models.

**Agreed, we removed the sentence the Reviewer is referring to.**

Table S1: Including standard deviation would be helpful for a reader to understand variability across the glacier.

**Table S1 does include the standard deviations (please see the screenshot from the Supplement previously submitted for the first round of review).**

**Table S1.** Minimum, maximum, and average values of the grounding zone width, ice thicknesses, bed slopes, and ice flow speed of TOT, MU, and REN glaciers calculated along the profiles of the corresponding glaciers.

| Glacier characteristics | | TOT | MU | REN | Data source |
|---|---|---|---|---|---|
| Grounding zone, km | Min | $1.2 \pm 0.4$ | $0.6 \pm 0.4$ | $1.3 \pm 0.4$ | Pairs of DInSAR interferograms |
| | Mean | $4.1 \pm 0.4$ | $2.1 \pm 0.4$ | $2.3 \pm 0.4$ | |
| | Max | $14.9 \pm 0.4$ | $5.1 \pm 0.4$ | $3.4 \pm 0.4$ | |
| Ice thickness, km | Min | $1.9 \pm 0.2$ | $2.0 \pm 0.2$ | $0.9 \pm 0.3$ | BedMachine2 (Morlighem et al., 2017) |
| | Mean | $2.2 \pm 0.1$ | $2.2 \pm 0.1$ | $1.1 \pm 0.2$ | |
| | Max | $2.4 \pm 0.2$ | $2.4 \pm 0.1$ | $1.2 \pm 0.1$ | |
| Bed slope, % | Min | $0.01 \pm 0.01$ | $0.2 \pm 0.2$ | $0.3 \pm 0.2$ | BedMachine2 (Morlighem et al., 2017) |
| | Mean | $1.2 \pm 0.1$ | $2.2 \pm 0.2$ | $1.4 \pm 0.2$ | |
| | Max | $4.0 \pm 0.2$ | $5.9 \pm 0.3$ | $3.5 \pm 0.3$ | |
| Ice flow speed, m / year | Min | $492 \pm 113$ | $171 \pm 41$ | $117 \pm 51$ | InSAR-based ice velocity data provided by MEaSUREs program (Rignot et al., 2017). |
| | Mean | $647 \pm 65$ | $335 \pm 20$ | $172 \pm 24$ | |
| | Max | $754 \pm 49$ | $381 \pm 18$ | $192 \pm 12$ | |

Figure S2: Why is only the lower boundary the smaller mesh size? Did you try a run with all small mesh sizes to see if there are different results? The monotonic decrease in GZ width with decrease in mesh size in Figure S1 is interesting. How do you know it evens out in the viscoelastic model? Wouldn't you expect scatter in GZ width once it is not mesh size dependent?

**Yes, we did run the model with small mesh sizes on the upper boundary and we did not find beneficial for two reasons:**

1. **Reducing the upper boundary mesh size does not improve model performance, which is expected since our primary focus is on the lower boundary. Adding more nodes to the upper boundary only increases the number of computed values in the upper part of the glacier, which are not relevant to our analysis of grounding lines.**
2. **Increasing the number of nodes on the upper boundary significantly increases computational time. This excessive use of computational resources is not justified, as it does not enhance accuracy.**

**We now mention our findings regarding the decreased upper-boundary mesh size (Supplement, lines 38-39)**

Importance of ice elasticity in simulating tide-induced grounding line variations along prograde bed slopes
*by Natalya Ross, Pietro Milillo, Kalyana Nakshatrala, Roberto Ballarini, Aaron Stubblefield, Luigi Dini*

**The monotonic decrease in grounding zone (GZ) width with decreasing mesh size, as shown in Figure S1, is an expected outcome—finer mesh resolution improves precision. This trend of decreasing an error when decreasing the mesh size has been well-documented in multiple finite element method (FEM) studies. Below, I list some FEM studies as an example:**

1. **Nakshatrala, K.B., Turner, D.Z., Hjelmstad, K.D. and Masud, A., 2006. A stabilized mixed finite element method for Darcy flow based on a multiscale decomposition of the solution. Computer Methods in Applied Mechanics and Engineering, 195(33-36), pp.4036-4049.**
2. **Joodat, S.H.S., Nakshatrala, K.B. and Ballarini, R., 2018. Modeling flow in porous media with double porosity/permeability: A stabilized mixed formulation, error analysis, and numerical solutions. *Computer Methods in Applied Mechanics and Engineering*, *337*, pp.632-676.**
3. **More, S.T. and Bindu, R.S., 2015. Effect of mesh size on finite element analysis of plate structure. *International Journal of Engineering Science and Innovative Technology*, *4*(3), pp.181-185.**
4. **Dutt, A., 2015. Effect of mesh size on finite element analysis of beam. *International Journal of Mechanical Engineering*, *2*(12), pp.8-10.**
5. **Strouboulis, T., Copps, K. and Babuška, I., 2001. The generalized finite element method. *Computer methods in applied mechanics and engineering*, *190*(32-33), pp.4081-4193.**
6. **Szabó, B.A., 1986. Mesh design for the p-version of the finite element method. *Computer Methods in Applied Mechanics and Engineering*, *55*(1-2), pp.181-197.**
7. **Strouboulis, T., Zhang, L. and Babuška, I., 2003. Generalized finite element method using mesh-based handbooks: application to problems in domains with many voids. *Computer Methods in Applied Mechanics and Engineering*, *192*(28-30), pp.3109-3161.**
8. **Strouboulis, T., Babuška, I. and Copps, K., 2000. The design and analysis of the generalized finite element method. *Computer methods in applied mechanics and engineering*, *181*(1-3), pp.43-69.**

Table S2: The r squared values of 1.0 are suspicious. How many data points are in the regression?
**For each bed slope and inflow speed, we approximate four modeled grounding zones, corresponding to the investigated glacier thicknesses of 1 km, 1.5 km, 2 km, and 2.5 km. Since we are approximating modeling results rather than observations, an R² value of 1 is not unrealistic; it simply indicates that, for the four considered thicknesses, the modeled relationship between grounding zone position and glacier thickness (for a given bed slope and inflow speed) is linear.**

Figure S4: Clarify if these are bar plots or histograms. If they are histograms, the bar width should encompass the full range of values in each bin. (e.g., In panel b, if the first bar is all profiles with a glacier thickness between 2.0 and 2.1 km, then it should stretch all the way to 2.1 km).
**We removed Figure S4, as we believe it is unnecessary for comprehension, given that it duplicates the information presented in Table S1.**

**We hope the revisions to the manuscript address the Reviewer's concerns and look forward to any further feedback.**